# Structures of neurexophilin–neurexin complexes reveal a regulatory mechanism of alternative splicing

Steven C Wilson[1] , K Ian White[1] , Qiangjun Zhou[1], Richard A Pfuetzner[1], Ucheor B Choi[1], Thomas C Südhof[1,2] & Axel T Brunger[1,2,*]

## Abstract

**Neurexins are presynaptic, cell-adhesion molecules that specify the functional properties of synapses via interactions with trans-synaptic ligands. Neurexins are extensively alternatively spliced at six canonical sites that regulate multifarious ligand interactions, but the structural mechanisms underlying alternative splicing-dependent neurexin regulation are largely unknown. Here, we determined high-resolution structures of the complex of neurexophilin-1 and the second laminin/neurexin/sex-hormone-binding globulin domain (LNS2) of neurexin-1 and examined how alternative splicing at splice site #2 (SS2) regulates the complex. Our data reveal a unique, extensive, neurexophilin–neurexin binding interface that extends the jelly-roll β-sandwich of LNS2 of neurexin-1 into neurexophilin-1. The SS2A insert of LNS2 augments this interface, increasing the binding affinity of LNS2 for neurexophilin-1. Taken together, our data reveal an unexpected architecture of neurexophilin–neurexin complexes that accounts for the modulation of binding by alternative splicing, which in turn regulates the competition of neurexophilin for neurexin binding with other ligands.**

**Keywords** alternative splicing; neurexin; neurexophilin; synapse; synaptic cell-adhesion molecules

**Subject Categories** Neuroscience; Structural Biology

**The EMBO Journal (2019) 38: e101603**

## Introduction

Brains process information via complex networks of neural circuits that govern behavior. Neural circuits are, in turn, composed of neurons that communicate with each other at synapses. The functional properties of these synapses regulate neural circuits and thereby contribute to the performance of all brain functions.

Neurexins (Nrxns) are a family of presynaptic cell-adhesion molecules that are important for specifying the functional properties of synapses (Südhof, 2017). Through their interactions with various binding partners, neurexins constitute part of a molecular code and act as hub molecules to organize and regulate synaptic function (Südhof, 2017).

Neurexins are encoded by three genes (*Nrxn1–3* in mice), each of which contains multiple promoters to drive expression of Nrxn1α–3α, Nrxn1β–3β, and Nrxn1γ. The extracellular sequences of α-neurexins contain six LNS domains (LNS1–LNS6) interspersed with epidermal growth factor-like domains (EGFA–EGFC), followed by a cysteine loop and a highly glycosylated stalk region (Fig 1A). The β-neurexins include only the C-terminal region of α-neurexins from the LNS6 domain onward. Nrxn1γ, in turn, lacks all LNS domains; its extracellular region contains only the C-terminal stalk region that is conserved in all neurexins (Sterky *et al*, 2017). All neurexins are embedded in the membrane by a single transmembrane region followed by a short (~55 residue) cytoplasmic tail. Remarkably, neurexins undergo extensive alternative splicing at six canonical sites (SS1–SS6; Fig 1A). As a result, they are expressed as thousands of isoforms in the brain (Ullrich *et al*, 1995; Schreiner *et al*, 2014; Treutlein *et al*, 2014). Alternative splicing of neurexins regulates the interactions of neurexins with a multitude of ligands, including neurexophilins (Nxphs), neuroligins (Nlgns), LRRTMs, cerebellins, and dystroglycan (Südhof, 2017).

Neurexophilins are a family of secreted, neuropeptide-like glycoproteins originally discovered as α-neurexin ligands and later found to bind to the LNS2 domain of α-neurexins (Petrenko *et al*, 1996; Missler *et al*, 1998; in the following we often refer to α-neurexins simply as neurexins or Nrxns). Neurexophilins are encoded by four genes (*Nxph1–4* in mice; Petrenko *et al*, 1996; Missler & Südhof, 1998; Missler *et al*, 1998), which are highly conserved in vertebrates but absent from invertebrates. The neurexophilins are unique in that they share no detectable homology with other known protein domains. Given the lack of similarity between neurexophilins and other neurexin ligands or neuropeptides, it is thus not surprising that the mechanism of the binding of neurexophilins to neurexins as

1   Department of Molecular and Cellular Physiology, Stanford University, Stanford, CA, USA
2   Howard Hughes Medical Institute, Stanford University, Stanford, CA, USA
   *Corresponding author. Tel: +1 650 736 1031; E-mail: brunger@stanford.edu

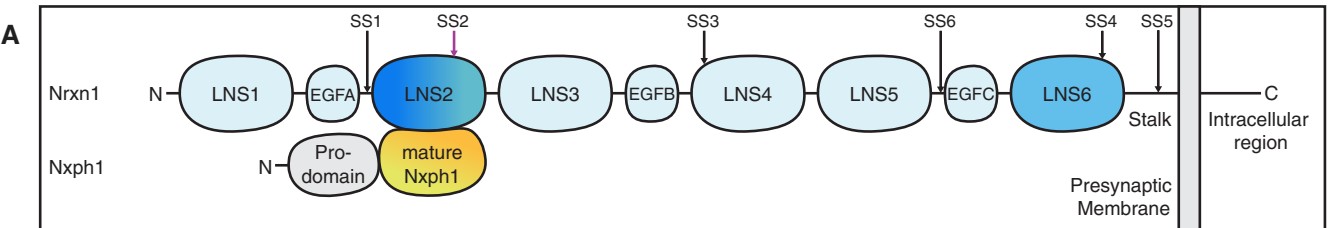

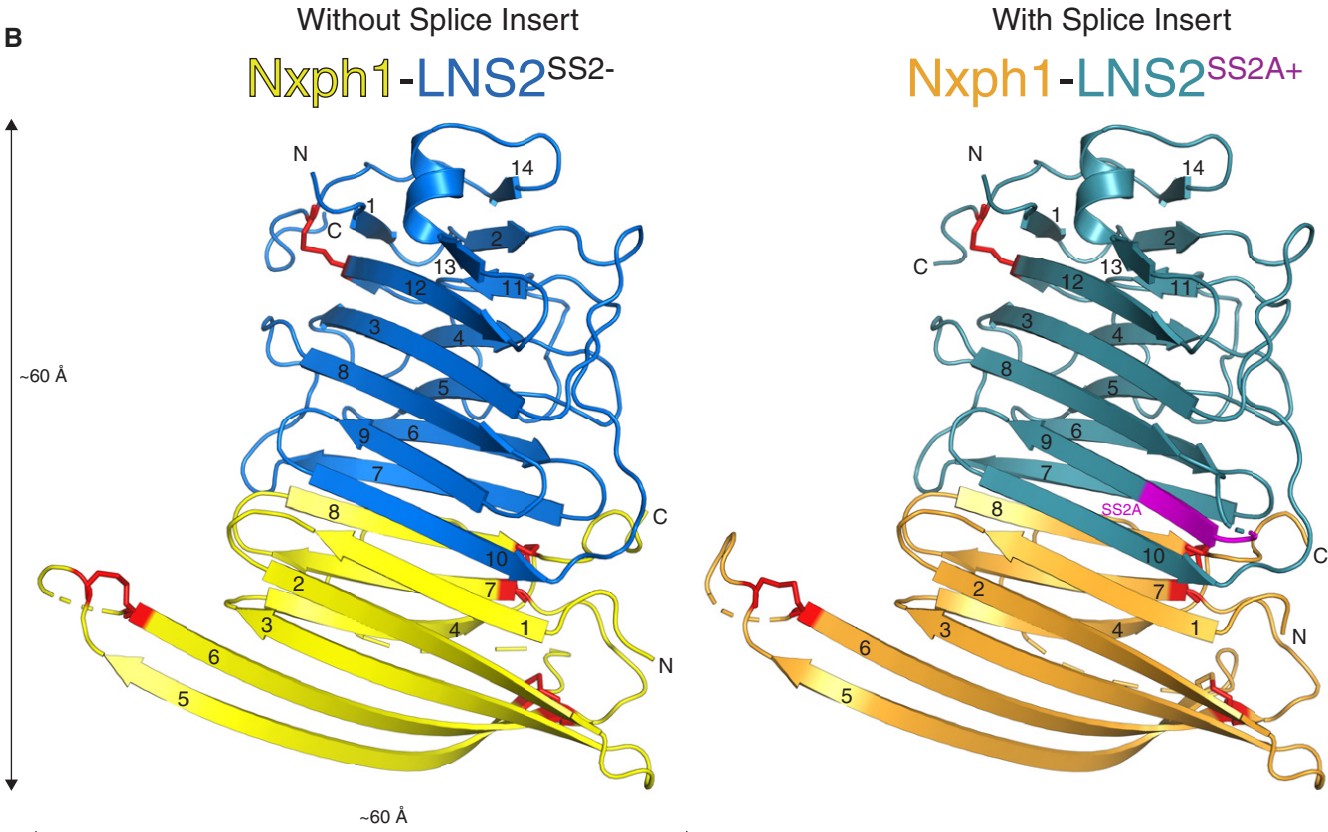

**Figure 1.  Domain structure of Nrxn1α and Nxph1 and overall architectures of the Nxph1-LNS2$^{SS2-}$ and Nxph1-LNS2$^{SS2A+}$ complexes.**

A   Schematic showing the domain structure of Nrxn1α and Nxph1 with approximate positions of alternative splice sites indicated by arrows. Colors of LNS2 and Nxph1 correspond to those in the structures in (B).

B   Crystal structures of Nxph1 in complex with Nrxn1 LNS2$^{SS2-}$ and LNS2$^{SS2A+}$. β-strands are numbered, and N- and C-termini are indicated. SS2A insert is shown in magenta. Disordered regions are shown as dashed lines, and disulfide bonds are shown as red sticks.

well as the regulation of this binding by alternative splicing remains unknown.

Different neurexophilin genes are expressed in distinct subpopulations of neurons in brain. For example, single-cell RNA-Seq studies showed that neurexophilin-1 (Nxph1) is abundantly produced in hippocampal inhibitory interneurons, Nxph3 in cortical excitatory neurons, and Nxph4 in inhibitory neurons of the hindbrain (Földy *et al*, 2016; Tasic *et al*, 2016; Saunders *et al*, 2018; Zeisel *et al*, 2018). The distinct expression profiles of neurexophilins resemble that of cerebellins throughout the brain, suggesting that they have cell type-specific roles (Seigneur & Südhof, 2017). Functionally, Nxph1 is the best-characterized neurexophilin, while little work has been done on the physiological roles of the other isoforms.

Electrophysiological recordings in brain slices from *Nxph1* KO mice revealed altered GABAergic synaptic transmission (Born *et al*, 2014), but the precise scope and mechanisms of Nxph1 function and the role of other neurexophilins remain to be examined.

Neurexophilins are composed of a signal peptide followed by a pro-domain, a polybasic cleavage site, and a mature fragment, thus resembling neuropeptides in that they are synthesized as a precursor protein and proteolytically processed to a mature form (Missler & Südhof, 1998; Fig 1A). The mature neurexophilin fragment was proposed to contain distinct N-terminal glycosylated and C-terminal cysteine-rich domains that are separated in Nxph4 only by a large loop sequence (Missler & Südhof, 1998). However, these putative N- and C-terminal domains of mature Nxph1, when individually fused

to Fc tags, were unable to co-precipitate LNS2, suggesting that the complete mature fragment of Nxph1 represents the minimal LNS2-binding region of Nxph1 (Reissner *et al*, 2014).

While alternative splicing of neurexins has dramatic effects on their interactions with ligands (Südhof, 2017), no structural data explain how alternative splicing regulates these interactions at the atomic level. For example, Nrxn1β lacking an insert in SS4 has a higher binding affinity for neuroligin-1 (Nlgn1) than Nrxn1β containing an insert in SS4 (Boucard *et al*, 2005; Comoletti *et al*, 2006; Shen *et al*, 2008; Elegheert *et al*, 2017), but the structural basis for this regulatory effect is unknown. The structure of the Nlgn1-Nrxn1β complex lacking the SS4 splice insert has been determined (Araç *et al*, 2007; Tanaka *et al*, 2011), but no structure is available for the Nlgn1-Nrxn1β complex containing the SS4 insert, even though most neuroligins bind to Nrxn1β containing an SS4 insert, albeit with a lower affinity. Moreover, no site of alternative splicing in neurexins except for SS4 has been studied in detail. Apart from SS4, only SS2 in the LNS2 domain is known to regulate binding of dystroglycan (Sugita *et al*, 2001). Because LNS2 is also the neurexin domain that binds to neurexophilins, dystroglycan competes with neurexophilins for binding to LNS2 (Reissner *et al*, 2014), but the significance of dystroglycan binding to neurexins remains unclear (Früh *et al*, 2016). Thus, there is an urgent need for insight into how neurexins interact with ligands such as neurexophilins in an alternative splicing-dependent manner via domains other than LNS6.

Here, we report splice-dependent interactions between Nxph1 and Nrxn1. Specifically, we determined high-resolution structures of Nxph1 in a complex with two Nrxn1 LNS2 SS2 splice variants and performed *in vitro* binding experiments to relate the structural data to binding properties. These structures reveal a previously unobserved type of protein–protein interface, wherein the β-sandwiches of Nxph1 and LNS2 form an extended, contiguous fold. Furthermore, the eight-amino acid splice insert of SS2 augments and stabilizes this interface, which results in increased binding affinity. Taken together, these data reveal a structural mechanism by which alternative splicing modulates the biochemical properties of a large and previously unobserved interface in a synaptic cell-adhesion protein complex.

## Results

### Structures of Nxph1 in complex with LNS2 splice variants of Nrxn1

Attempts at producing Nxph1 using insect cell expression systems failed due to intracellular retention of the protein, so the BacMam system was used to co-express secreted mature Nxph1 and Nrxn1 LNS2 proteins from HEK293S GnTI- cells (Dukkipati *et al*, 2008; Goehring *et al*, 2014). This led to the efficient secretion of Nxph1-Nrxn1 LNS2 complexes. Two splice variants of Nrxn1—LNS2$^{SS2-}$ with no insert in SS2 and LNS2$^{SS2A+}$ with a conserved eight-residue insert in SS2—were used. Highly homogeneous samples of the Nxph1-Nrxn1 LNS2$^{SS2-}$ and Nxph1-Nrxn1 LNS2$^{SS2A+}$ complexes with 1:1 stoichiometry were prepared as assessed by size-exclusion chromatography–multi-angle light scattering (SEC-MALS; Fig EV1C and D). In the following, we refer to these complexes simply as the Nxph1-LNS2$^{SS2-}$ and Nxph1-LNS2$^{SS2A+}$ complexes.

Nxph1 has *N*-linked glycans at amino acid positions N146, N156, and N162, while LNS2 is not glycosylated. Crystallization trials using samples of high mannose-glycosylated Nxph1-LNS2 complex did not yield crystals. EndoH or EndoF1 deglycosylated proteins also could not be crystallized, probably because the glycosidases did not completely deglycosylate Nxph1 as evaluated by SDS–PAGE. Therefore, in order to crystallize Nxph1-LNS2 complexes, glycosylation of Nxph1 was prevented by mutating the aforementioned *N*-linked glycosylation sites to aspartic acid (Nxph1$^{3ND}$). Crystallization trials of Nxph1$^{3ND}$-LNS2$^{SS2-}$ and Nxph1$^{3ND}$-LNS2$^{SS2A+}$ complexes yielded crystals that diffracted to 1.9 Å resolution.

Because the structure of the LNS2$^{SS2-}$ domain of Nrxn1 was known, the structure of the Nxph1-LNS2$^{SS2-}$ complex was determined by molecular replacement with the Nrxn1 LNS2$^{SS2-}$ domain as the search model, resulting in a well-resolved electron density map for the unknown structure of Nxph1—roughly half of the amino acid content of the structure of the complex. This electron density map was of sufficient quality to build an atomic model of Nxph1 by automated methods (Materials and Methods; Table 1). In turn, the structure of this complex was used as a search model for determining the structure of the Nxph1-LNS2$^{SS2A+}$ complex by molecular replacement.

Both the Nxph1-LNS2$^{SS2-}$ and Nxph1-LNS2$^{SS2A+}$ structures exhibit a similar overall architecture, with Nxph1 forming a striking extension of the LNS2 jelly-roll β-sandwich described previously (Sheckler *et al*, 2006; Chen *et al*, 2011; Miller *et al*, 2011; Fig 1B). Overall, the LNS2 structure consists of 14 anti-parallel β-strands, one α-helix, and one disulfide bond connecting C444 to C480 (Fig EV2). The structure of the LNS2 domain in both Nxph1-LNS2 complexes is similar to those of LNS2 alone (PDB IDs 2H0B, 3R05, and 3POY) except for the β$_7$ and β$_{10}$ strands and the loop that connects β$_{10}$ to the rest of the molecule (Appendix Fig S1). β$_7$ and β$_{10}$ form part of the interface to Nxph1. Thus, complex formation with Nxph1 reorders these β-strands and the loop.

The Nxph1 structure comprises eight β-strands and contains three disulfide bonds near loop regions. The disulfide bonds in Nxph1 link cysteine pairs C194-C231, C210-C218, and C239-C256 (Figs 1B and EV3). The connectivity pattern of the disulfide bonds in the Nxph1 structure is consistent with the pattern inferred by mass spectrometry (Reissner *et al*, 2014). The β-sandwich of Nxph1 consists of two anti-parallel β-sheets formed by β$_1$–β$_3$, β$_4$, β$_7$, and β$_8$, while β$_5$ and β$_6$ run anti-parallel to one another and are solvent-exposed (Fig 1B). The single β-sandwich architecture of the Nxph1 structure differs markedly from the previously proposed architecture that predicted the presence of distinct N- and C-terminal domains connected by a linker (Missler & Südhof, 1998). Rather, the β-strands formed by these N- and C-terminal regions are interleaved in the structure, mapping to the strands β$_1$–β$_4$ and β$_5$–β$_8$ in the Nxph1 structure, respectively (Appendix Fig S2). The loop connecting β$_4$ to β$_5$ contains a ~50 glycine- and proline-rich amino acid sequence in Nxph4 but not in the other neurexophilins (Appendix Fig S2). Analysis of the Nxph1 structure using DALI (Holm & Laakso, 2016) revealed that Nxph1 shares only very low-level structural similarity with a few proteins. Among these, the highest scoring structure is a possible leukocidin subunit (*z*-score = 8.9, sequence identity = 10%, RMSD = 3.2) and the only proteins that have similarity to the Nxph1 topology are members of the adaptin family (*z*-score = 7.9, sequence identity = 4%, RMSD = 3.3).

**Table 1.** Data collection and refinement statistics.

| Structure | Nxph1-Nrxn1 LNS2$^{SS2-}$ | Nxph1-Nrxn1 LNS2$^{SS2A+}$ |
|---|---|---|
| Beamline | APS NE-CAT 24-ID-C | SSRL 12-2 |
| Wavelength | 0.97910 Å | 0.97946 Å |
| Resolution range | 45.60–1.94 (2.01–1.94) | 45.92–1.95 (2.02–1.95) |
| Space group | C 1 2 1 | C 1 2 1 |
| Unit cell | 71.7 61.0 79.6 90 106.7 90 | 72.0 61.4 79.4 90 106.2 90 |
| Total reflections | 317,952 (21,624) | 279,800 (12,212) |
| Unique reflections | 24,370 (1,662) | 23,404 (583) |
| Multiplicity | 13.0 (9.1) | 12.0 (6.3) |
| Completeness (%) | 94.99 (68.58) | 83.17 (24.18) |
| Mean I/sigma (I) | 14.05 (1.29) | 11.40 (0.78) |
| Wilson B-factor | 39.50 | 20.68 |
| R-merge | 0.100 (1.214) | 0.108 (1.421) |
| CC1/2 | 0.999 (0.796) | 0.998 (0.559) |
| Reflections used in refinement | 23,258 (1,657) | 20,360 (583) |
| Reflections used for R-free | 1,992 (142) | 1,997 (57) |
| R-work | 0.198 (0.288) | 0.196 (0.314) |
| R-free | 0.247 (0.331) | 0.240 (0.315) |
| Number of non-hydrogen atoms | 2,656 | 2,791 |
| Macromolecules | 2,589 | 2,611 |
| Solvent | 67 | 180 |
| Protein residues | 325 | 327 |
| RMS (bonds) | 0.009 | 0.005 |
| RMS (angles) | 0.88 | 1.07 |
| Ramachandran favored (%) | 97.16 | 95.58 |
| Ramachandran allowed (%) | 2.84 | 4.42 |
| Ramachandran outliers (%) | 0 | 0 |
| Rotamer outliers (%) | 2.74 | 0 |
| Clashscore | 6.84 | 5.42 |
| Average B-factor | 76.06 | 35.86 |
| Macromolecules | 76.47 | 35.83 |
| Solvent | 59.94 | 36.28 |
| Number of TLS groups | 2 | 2 |

Statistics for the highest resolution shell are shown in parentheses.

The heterodimeric interfaces between the Nxph1 and LNS2 β-sandwiches in the Nxph1-LNS2$^{SS2-}$ and Nxph1-LNS2$^{SS2A+}$ structures have buried surface areas of 1,223 Å$^2$ and 1,209 Å$^2$, respectively, much larger than another interface formed between Nxph1 molecules in the crystal lattice (Appendix Fig S3A and B). These interaction interfaces of Nxph1 may be involved in homomeric interactions, as suggested by the presence of dimers and tetramers of Nxph1 alone (based on SEC-MALS and SEC-SAXS experiments; Figs EV1E and EV4A; Table EV1).

The Nxph1-LNS2 interface in both complexes is formed via antiparallel β-sheet extensions mediated by interactions between β$_1$ and β$_8$ of Nxph1 with β$_{10}$ and β$_7$ of LNS2, respectively (Fig 1B). Furthermore, Nxph1 and LNS2 interlock in a staggered fashion with hydrophobic and polar interactions from Nxph1 β$_8$ and LNS2 β$_{10}$ spanning the β-sandwich formed by Nxph1 and LNS2 (Appendix Fig S3C and D). A PISA search using default parameters identified no instances of similar architecture within all deposited structures in the Protein Data Bank (RCSB). Thus, this is a striking example of a heteromeric anti-parallel β-sandwich extension.

## Key elements of the Nxph1-LNS2$^{SS2-}$ interface

Visual inspection of the Nxph1-LNS2$^{SS2-}$ structure revealed multiple key interface residues (Fig 2A). Among these residues, I401 of LNS2$^{SS2-}$ interacts with a particularly deep hydrophobic binding pocket in Nxph1 (formed primarily by Y249 and L251; Fig 2A and

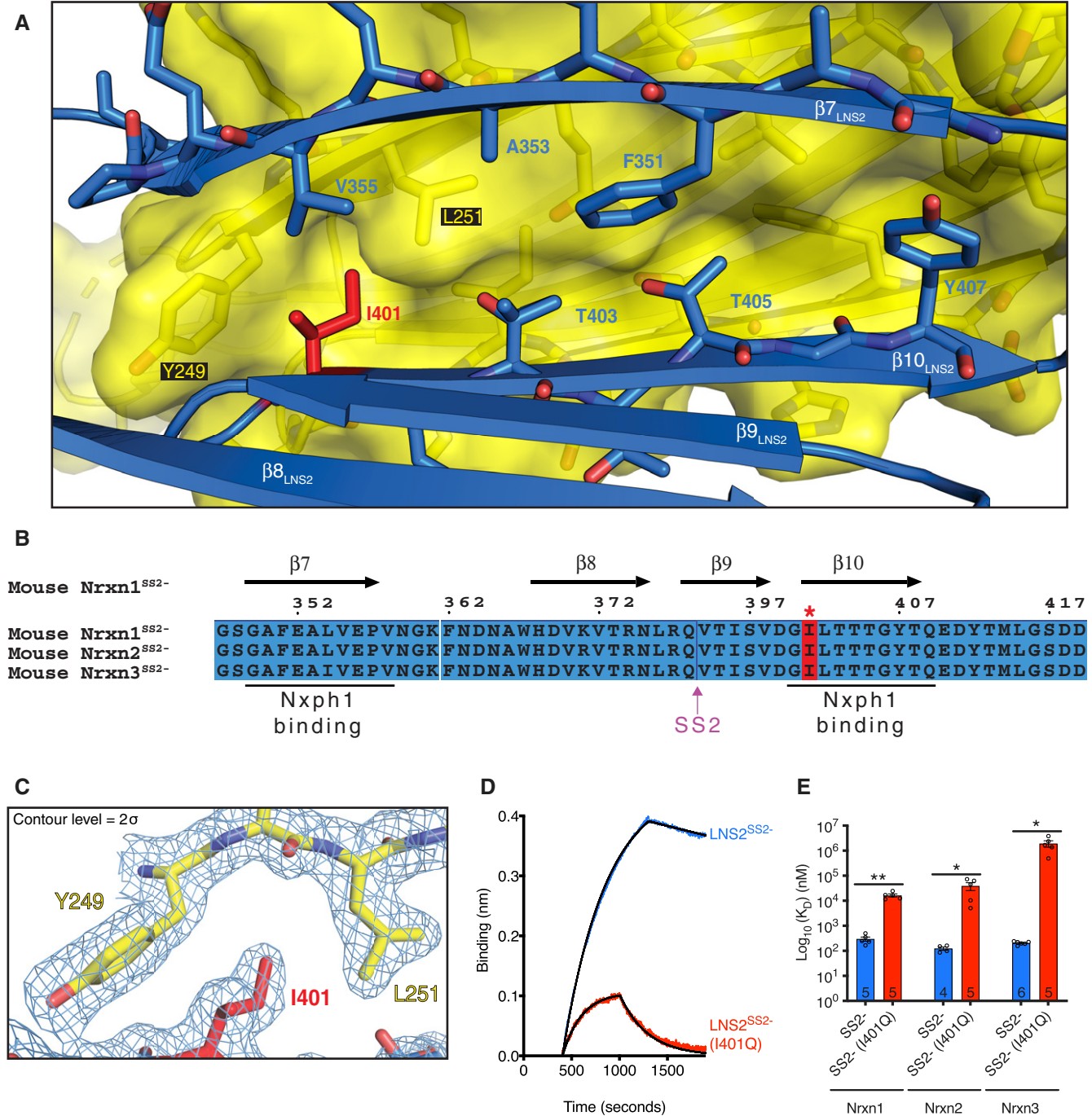

**Figure 2. Key features of the Nxph1-LNS2$^{SS2-}$ interface.**

A   View of the Nxph1-LNS2$^{SS2-}$ interface showing key residues as sticks. The yellow surface represents the Nxph1 molecule.

B   Sequence alignment of mouse Nrxn1–3 LNS2$^{SS2-}$ showing key regions involved in Nxph1 binding. Sequences are numbered in reference to the canonical UniProt mouse Nrxn1 sequence Q9CS84-1. Secondary structure elements of LNS2$^{SS2-}$ are shown at the top and correspond to elements shown in (A).

C   Electron density map showing the LNS2$^{SS2-}$ I401 residue (highlighted in red in B) interacting with a hydrophobic binding pocket formed primarily by Y249 and L251 of Nxph1.

D   Representative BLI traces showing decreased binding of Nxph1 to the Nrxn1 LNS2$^{SS2-}$ (I401Q) mutant compared to that of wild-type Nrxn1 LNS2$^{SS2-}$. Replicate numbers are shown in (E).

E   Comparison of binding affinities for Nxph1 binding to mouse Nrxn1–3 LNS2$^{SS2-}$ and LNS2$^{SS2-}$ (I401Q), plotted on a logarithmic scale. Error bars represent SEM, and replicate numbers are indicated in bars. Significance values were calculated using Welch's $t$-test, *$P < 0.05$, **$P < 0.01$.

C). These interface residues are highly conserved across both mouse (Fig 2B) and vertebrate neurexins (Fig EV2) as well as neurexophilins (Fig EV3). However, these Nxph-binding sites are not conserved in invertebrate neurexins, consistent with the lack of neurexophilins in invertebrates (Fig EV2).

To validate the interface observed in the Nxph1-LNS2$^{SS2-}$ crystal structure, mutagenesis and co-immunoprecipitation experiments were performed for selected neurexin residues at the interface (Fig 2A). The residues were selected by visual inspection of the structure, along with energetic analyses of the interacting residues by PISA. Mutagenesis of Nrxn3 LNS2$^{SS2-}$ residues, corresponding to I401, T405, and Y407 of Nrxn1 LNS2$^{SS2-}$, resulted in decreased co-immunoprecipitation of LNS2$^{SS2-}$ with Nxph1 (Appendix Fig S4A). These residues are all located in the LNS2$^{SS2-}$ $\beta_{10}$ strand that is part of the hydrophobic core of the Nxph1-LNS2$^{SS2-}$ interface. Individual mutation of these residues to glutamine substantially reduced Nxph1 co-immunoprecipitation. Of these mutants, I401Q had the greatest effect on Nxph1 binding to LNS2$^{SS2-}$, nearly eliminating the interaction (Appendix Fig S4A). Similarly, confocal imaging experiments showed that the I401Q mutation reduced localization of Nxph1 with membrane-bound Nrxn1 LNS2$^{SS2-}$ when co-expressed in HEK293T cells (Fig EV5A). Interestingly, this mutation was previously found to have an effect in experiments using scanning mutagenesis and imaging of co-expressed constructs to explore the binding of Nxph1 to LNS2 (Reissner et al, 2014; Neupert et al, 2015).

Biolayer Interferometry (BLI) was then used to quantitatively assess the strength of the Nxph1-LNS2 interaction (Fig 2D and E; Table 2). Nxph1 binds Nrxn1 LNS2$^{SS2-}$ with nanomolar affinity (Fig 2E; Table 2). While glycosylation of Nxph1 may stabilize its binding to neurexin (Reissner et al, 2014), we were unable to produce well-behaved Nxph1 without glycans on its own and thus could not assess the effects of glycosylation on Nxph1-neurexin interactions. The I401Q mutation in Nrxn1 LNS2$^{SS2-}$ greatly decreased binding of Nxph1 to Nrxn1–3 LNS2$^{SS2-}$ (Figs 2D and E, and EV5B; Table 2). The mutant proteins used in the binding experiments were folded as assessed by circular dichroism (CD) spectroscopy (Appendix Fig S4B), so the reduction in binding affinity is not due to a folding defect. Furthermore, the same mutant proteins exhibited monodisperse size-exclusion chromatography elution profiles as evaluated by SEC-MALS experiments with the Nrxn2 LNS2$^{SS2-}$ (I401Q) domain (Fig EV1H). Thus, the reduction in binding affinity is not due to aggregation. The primary sequence conservation of the interface residues suggests that the Nxph1-LNS2$^{SS2-}$ binding interface is maintained in all vertebrates (Figs EV2 and EV3).

## Structural changes and modulation of binding by alternative splicing

In the structure of the Nxph1-LNS2$^{SS2A+}$ complex, the 384-HAM-386 sequence of SS2A extends the N-terminal part of LNS2 $\beta_9$ (Fig 3A). The other part of the SS2A sequence (379-HSGIG-383) forms a disordered, solvent-exposed loop as suggested by corresponding weak electron density.

Structural remodeling at and near the splice insert site as a function of SS2 alternative splicing is striking and extensive. Difference density maps calculated by subtracting the observed structure factor amplitudes of the Nxph1-LNS2$^{SS2-}$ structure from those of the Nxph1-LNS2$^{SS2A+}$ structure showed strong positive electron density throughout the region near SS2A (Fig 3B). Strong positive difference peaks revealed the extension of the $\beta_9$ 384-HAM-386 part of the SS2A insert sequence in the Nxph1-LNS2$^{SS2A+}$

**Table 2. Biolayer Interferometry binding data.**

| Binding pair | Mean $K_D$ (nM) | $K_{on}$ (1/M*s) | $K_{off}$ (1/s) |
|---|---|---|---|
| Nxph1-Nrxn1 LNS2$^{SS2-}$ | 299.2 ± 58.3 | 398.6 ± 31.5 | 1.2E-04 ± 2.3E-05 |
| Nxph1-Nrxn1 LNS2$^{SS2-}$ (I401Q) | 16,420 ± 2,299 | 219.6 ± 22.0 | 3.4E-03 ± 1.8E-04 |
| Nxph1-Nrxn1 LNS2$^{SS2A+}$ | 49.0 ± 8.4 | 2,043 ± 139.6 | 9.7E-05 ± 1.2E-05 |
| Nxph1-Nrxn1 LNS2$^{SS2A+}$ (I401Q) | 472.0 ± 66.4 | 419.2 ± 27.8 | 2.0E-04 ± 3.3E-05 |
| Nxph1-Nrxn1 LNS2$^{SS2AB+}$ | 43.3 ± 4.7 | 2,168 ± 2.5 | 9.4E-05 ± 1.0E-05 |
| Nxph1-Nrxn1 LNS2$^{SS2AB+}$ (I401Q) | 499.3 ± 17.7 | 673.8 ± 23.5 | 3.4E-04 ± 5.5E-06 |
| Nxph1-Nrxn2 LNS2$^{SS2-}$ | 124.3 ± 18.7 | 553.5 ± 13.5 | 6.9E-05 ± 1.1E-05 |
| Nxph1-Nrxn2 LNS2$^{SS2-}$ (I401Q) | 39,340 ± 13,662 | 380.8 ± 25.3 | 2.5E-03 ± 2.1E-04 |
| Nxph1-Nrxn3 LNS2$^{SS2-}$ | 206.5 ± 12.2 | 444.5 ± 7.5 | 9.2E-05 ± 5.9E-06 |
| Nxph1-Nrxn3 LNS2$^{SS2-}$ (I401Q) | 1,941,800 ± 555,480 | 10.5 ± 4.6 | 1.2E-02 ± 1.1E-03 |
| Nxph1-Nrxn3 LNS2$^{SS2A+}$ | 31.4 ± 3.5 | 2,060 ± 108.2 | 6.4E-05 ± 7.2E-06 |
| Nxph3-Nrxn1 LNS2$^{SS2-}$ | 260.0 ± 83.8 | 209.3 ± 14.6 | 1.0E-04 ± 4.1E-05 |
| Nxph3-Nrxn2 LNS2$^{SS2-}$ | 212.2 ± 52.0 | 301.8 ± 14.4 | 6.7E-05 ± 1.9E-05 |
| Nxph3-Nrxn3 LNS2$^{SS2-}$ | 177.3 ± 18.1 | 178.5 ± 10.0 | 3.1E-05 ± 3.3E-06 |
| Nxph3-Nrxn1 LNS2$^{SS2-}$ (I401Q) | 3,345 ± 276.3 | 193.8 ± 5.5 | 6.4E-04 ± 3.7E-05 |
| Nxph3-Nrxn2 LNS2$^{SS2-}$ (I401Q) | 1,315 ± 143.2 | 350.3 ± 37.4 | 4.4E-04 ± 2.1E-06 |
| Nxph3-Nrxn3 LNS2$^{SS2-}$ (I401Q) | 3,501 ± 213.6 | 153.0 ± 11.1 | 5.3E-04 ± 1.6E-05 |

Error values are SEM.

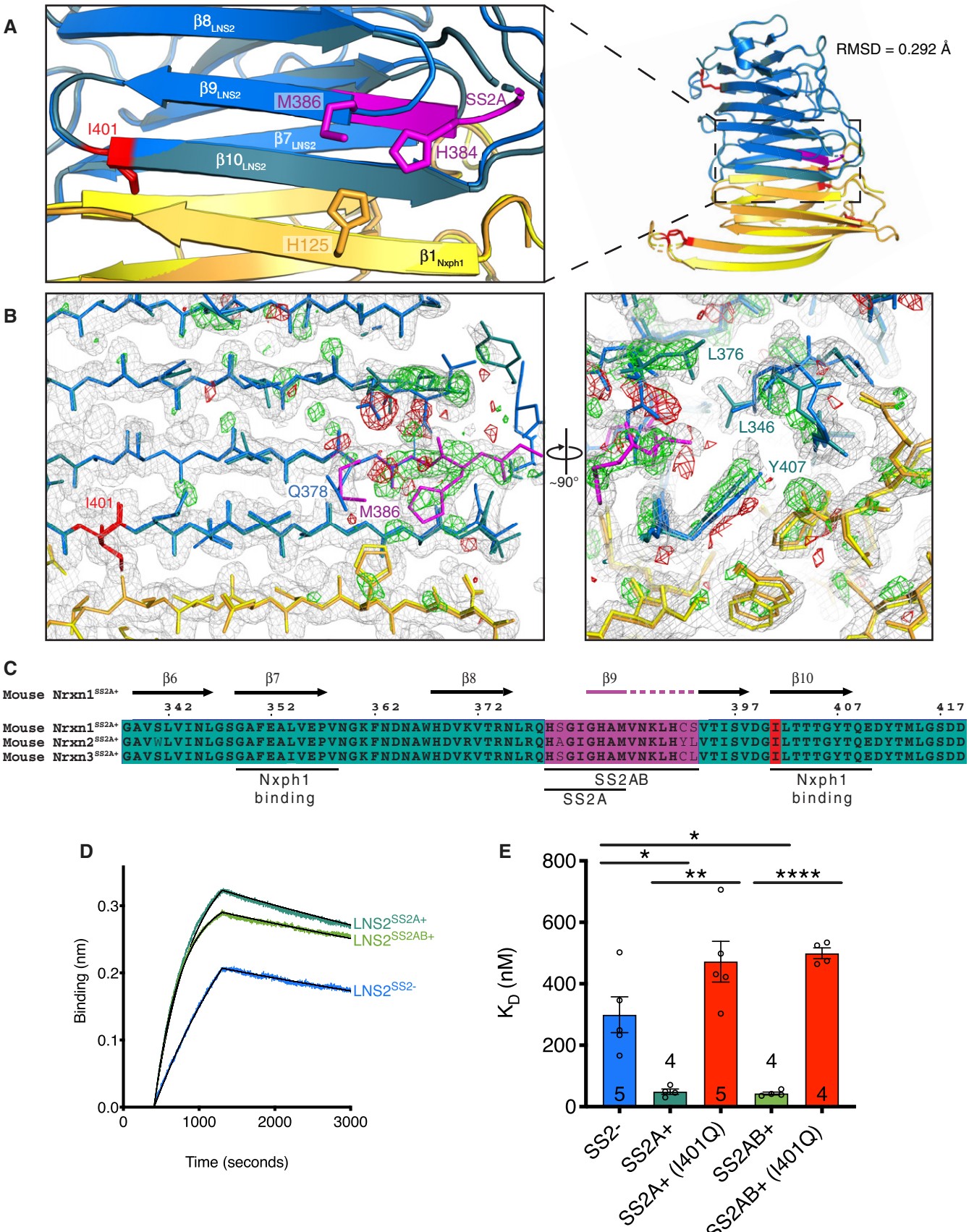

Figure 3.

◄

**Figure 3. Comparison of the Nxph1-LNS2$^{SS2A+}$ structure to the Nxph1-LNS2$^{SS2-}$ structure and modulation of Nxph1-Nrxn1 LNS2 binding by alternative splicing.**

A Overall view and close-up of superimposed Nxph1-LNS2$^{SS2-}$ and Nxph1-LNS2$^{SS2A+}$ structures in cartoon representation.

B Electron density maps comparing the Nxph1-LNS2$^{SS2-}$ and Nxph1-LNS2$^{SS2A+}$ structures. These maps cover the same region as shown in the close-up in (A). The 2mF$_o$-DF$_c$ map for the Nxph1-LNS2$^{SS2A+}$ structure is shown in gray at a contour level of 1 σ. The positive and negative F$_o$–F$_o$ difference density maps (the difference between Nxph1-LNS2$^{SS2-}$ and Nxph1-LNS2$^{SS2A+}$ structure factor amplitudes using phases from the Nxph1-LNS2$^{SS2A+}$ model) are shown in green and red, respectively, at a contour level of 3 σ.

C Sequence alignment of mouse Nrxn1–3 LNS2$^{SS2A/AB+}$ showing key regions involved in Nxph1 binding and SS2A and SS2AB inserts. Sequences are numbered in reference to the canonical UniProt mouse Nrxn1 sequence Q9CS84-1. Secondary structure elements are shown at the top and correspond to elements shown in the Nxph1-LNS2$^{SS2A+}$ structure in (A).

D Representative BLI traces showing increased binding of Nxph1 to Nrxn1 LNS2$^{SS2A+}$ and Nrxn1 LNS2$^{SS2AB+}$ compared to that of Nrxn1 LNS2$^{SS2-}$. Replicate numbers are shown in (E).

E Comparison of binding affinities for Nxph1 binding to mouse Nrxn1 LNS2 splice variants and splice insert containing Nrxn1 LNS2 with the I401Q mutation; error bars represent SEM, and replicate numbers are indicated in or above bars. Significance values were calculated using Welch's *t*-test, *$P < 0.05$, **$P < 0.01$, ****$P < 0.0001$.

structure. Along with this extension, a variety of secondary rearrangements is evident in the difference density map. For example, the electron density representative of Q378 in the Nxph1-LNS2$^{SS2-}$ structure is replaced by that of M386 in the Nxph1-LNS2$^{SS2A+}$ structure. Furthermore, the M386 and H384 residues of SS2A extend across β$_{10}$ of LNS2 to interact with H125 on β$_1$ of Nxph1. The H384 and M386 residues have buried surface areas upon complex formation of approximately 13 Å$^2$ and 12 Å$^2$, respectively, as calculated by PISA. The presence of SS2A also likely stabilizes the region proximal to the splice insert by the addition of hydrophobic interactions. Specifically, in the Nxph1-LNS2$^{SS2A+}$ structure, but not the Nxph1-LNS2$^{SS2-}$ structure, L376 points into the core of the molecule toward L346, which shifts along with the backbone of β$_7$ further into the core of the molecule. In addition, there is a shift of Y407 into the core of the LNS2 molecule in the Nxph1-LNS2$^{SS2A+}$ structure. Taken together, these structures show

that SS2A adds specific inter-molecular interactions to the Nxph1-LNS2 interface while also inducing regional conformational changes to stabilize the complex.

Given the extensive effect of the SS2A insert on the Nxph1-LNS2 structure and the proximity of SS2 to the Nxph1-binding site in LNS2 (Figs 1A and 3A–C, and EV2), inserts in SS2 are expected to increase the affinity of Nxph1 for LNS2. Indeed, Nxph1 binds to Nrxn1 LNS2$^{SS2A+}$ with ~6-fold higher affinity than for Nrxn1 LNS2$^{SS2-}$ (Fig 3D and E; Table 2). In these analyses, greater K$_{on}$ and nearly identical K$_{off}$ values were observed for Nxph1 binding to Nrxn1 LNS2$^{SS2A+}$ (Table 2), explaining the binding affinity differences (Fig 3D).

In addition to the SS2A insert found in all neurexins, a less frequent splice variant includes a larger insert extended by seven additional residues: SS2AB (Ullrich *et al*, 1995) (Figs 3C and EV2). Using BLI, Nrxn1 LNS2 containing this SS2AB insert, LNS2$^{SS2AB+}$,

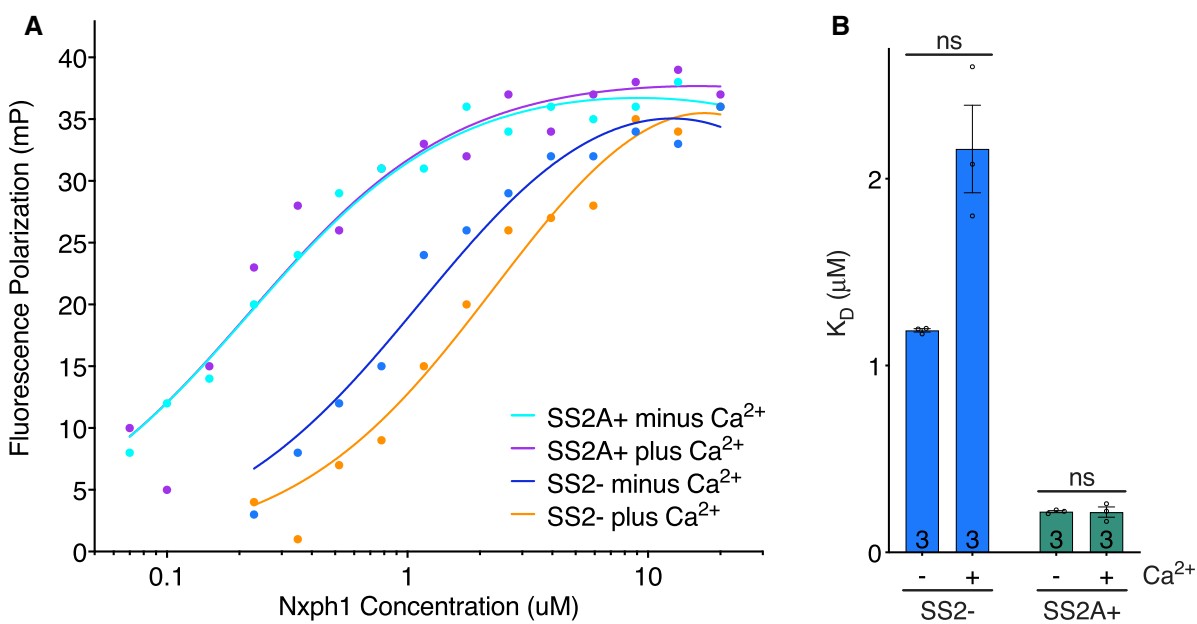

**Figure 4. The SS2A insert does not affect Ca$^{2+}$ sensitivity of Nxph1-LNS2 interactions.**

A Representative fluorescence polarization binding curves are shown for LNS2$^{SS2-}$ and LNS2$^{SS2A+}$ interacting with Nxph1 with or without Ca$^{2+}$ added. Replicate numbers are shown in (B).

B Comparison of binding affinities of Nxph1 for Nrxn1 LNS2$^{SS2-}$ and Nrxn1 LNS2$^{SS2A+}$ with or without Ca$^{2+}$ added; replicate numbers are indicated in bars. Error bars represent SEM, and significance values were calculated using Welch's *t*-test. ns, not significant.

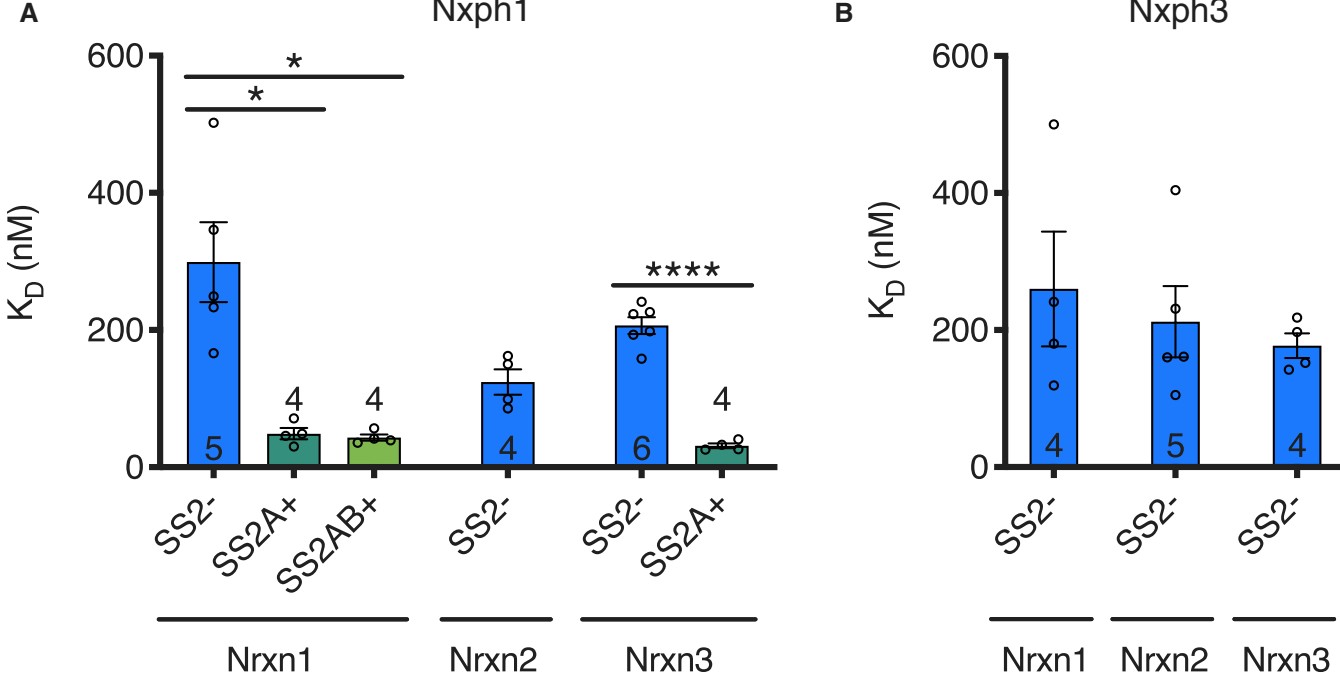

**Figure 5.   Isoform specificity of neurexin–neurexophilin interactions.**
A   Comparison of binding affinities of Nxph1 binding to different Nrxn1–3 LNS2 isoforms.
B   Comparison of binding affinities for Nxph3 binding to Nrxn1–3 LNS2$^{SS2-}$.

Data information: Error bars show SEM, and replicate numbers are indicated in or above bars. Significance values were calculated using Welch's *t*-test, $^*P < 0.05$, $^{****}P < 0.0001$.

binds Nxph1 with similar kinetics and affinity as LNS2$^{SS2A+}$ (Fig 3D and E; Table 2).

Despite their differences, many amino acid interactions are nevertheless conserved in the Nxph1-LNS2$^{SS2A+}$ and Nxph1-LNS2$^{SS2-}$ structures. For example, the Nxph1-LNS2$^{SS2A+}$ interface contains the same key I401 residue found interdigitating with Nxph1 at the Nxph1-LNS2$^{SS2-}$ interface (Fig 3A, Appendix Fig S3D). Binding of Nxph1 to LNS2$^{SS2A+}$ and LNS2$^{SS2AB+}$ is similarly affected by the I401Q mutation in these proteins. In both Nrxn1 LNS2 splice variants, the I401Q mutations also significantly reduced binding of Nxph1 (Fig 3E). Again, these mutant proteins are folded as assessed by CD spectroscopy (Appendix Fig S4B) and exhibited monodisperse size-exclusion chromatography elution profiles, ensuring that the reduction in binding affinity was not due to a folding defect or to aggregation. The effects of the I401Q mutation in Nrxn1 LNS2$^{SS2A+}$ and in Nrxn1 LNS2$^{SS2AB+}$ on Nxph1 binding are less pronounced than those observed for the I401Q mutation in the splice insert-free Nrxn1 LNS2$^{SS2-}$ variant, consistent with the observed stabilization of the Nxph1-Nrxn1 interface via additional interactions contributed by the SS2A insert.

### The SS2 insert has no effect on the calcium sensitivity of Nxph1-LNS2 interactions

We did not observe electron density in the Ca$^{2+}$ binding region of LNS2 (Sheckler *et al*, 2006), presumably because Ca$^{2+}$ was not included during crystallization. Considering the proximity of the Nxph1-LNS2 interface to the LNS2 Ca$^{2+}$-binding region, we tested the effect of Ca$^{2+}$ on Nxph1-LNS2 interactions. Using a fluorescence anisotropy binding assay (Fig 4A), 2 mM Ca$^{2+}$ had no effect on the binding affinities of Nxph1 for Nrxn1 LNS2$^{SS2A+}$ or for Nrxn1 LNS2$^{SS2-}$ (Fig 4B). In addition, inclusion of 100 μM EDTA in Ca$^{2+}$-free fluorescence anisotropy binding assays had no effect on the binding affinities, ruling out the possibility that trace amounts of Ca$^{2+}$ in the buffer could affect the Nxph1-LNS2 interactions (Appendix Fig S5). Thus, Ca$^{2+}$ is not a major regulator of the Nxph1-Nrxn1 interaction, consistent with the notion that extracellular Ca$^{2+}$ is not thought to be a regulatory parameter.

### Isoform specificity of neurexophilin–neurexin interactions

In addition to investigating the interactions of Nxph1 with Nrxn1, the binding of Nxph1 to mouse Nrxn2 and Nrxn3 was assessed. Nxph1 binds to the LNS2$^{SS2-}$ domains of all neurexins (Nrxn1, Nrxn2, and Nrxn3) with similar nanomolar affinities (Fig 5A; Table 2), consistent with the sequence conservation of neurexin residues at the binding interface (Figs 2B and EV2). Nrxn3 LNS2$^{SS2A+}$ again binds Nxph1 with a ~6-fold higher affinity than Nrxn3 LNS2$^{SS2-}$ (Fig 5A; Table 2). As with Nrxn1 binding to Nxph1, greater K$_{on}$ and nearly identical K$_{off}$ values were observed for Nrxn3 LNS2$^{SS2A+}$ binding to Nxph1 (Table 2). The SS2A-induced increase in Nrxn3 binding affinity for Nxph1 is similar to that of Nrxn1, consistent with the high primary sequence conservation of the splice insert (Fig 3C). We did not test the other splice variants of Nrxn2, and the Nrxn3 LNS2$^{SS2AB+}$ variant did not express well, preventing further studies.

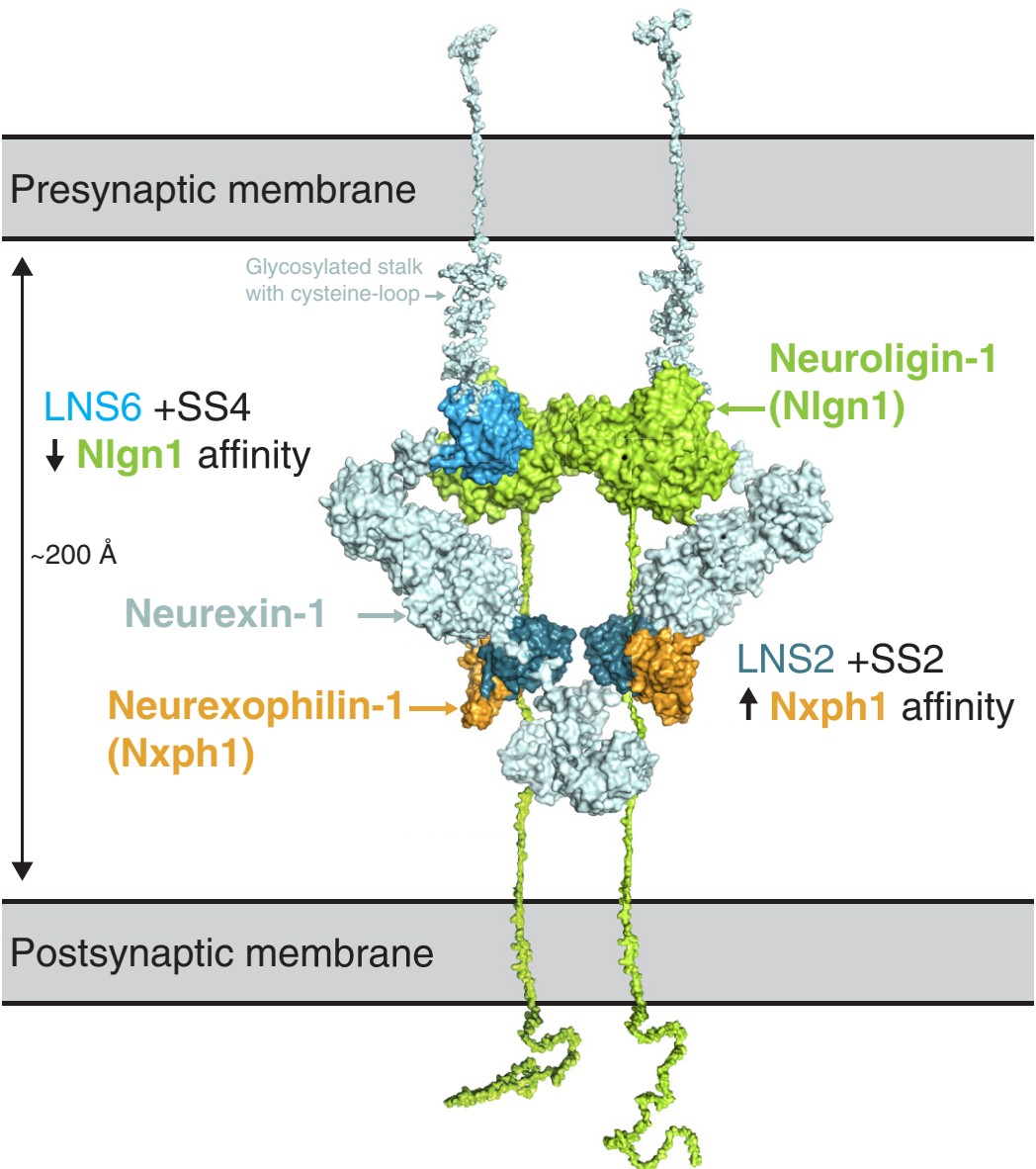

**Figure 6. Model of neurexin bound to both neurexophilin and neuroligin in the synapse.**

A hypothetical composite structure of Nrxn1 bound to both Nxph1 and Nlgn1 is shown. The Nxph1-LNS2$^{SS2A+}$ complex structure was superimposed on the near full-length crystal structure of Nrxn1α (3R05) containing LNS2-LNS6 domains (Chen *et al*, 2011). Note that there are no coordinates for the Nrxn1α LNS1 or EGFA domains; structures shown here are models based on the EGFC and LNS6 domains of Nrxn1α (3R05). Nlgn1 from the crystal structure of the Nrxn1β-Nlgn1 complex (3BIW; Araç *et al*, 2007) is modeled by superimposition of that complex with the structure of Nrxn1α. Stalk regions, transmembrane domains, and intracellular regions for Nlgn1 and Nrxn1α were modeled based on mouse protein sequences using PyMOL. For Nrxn1, the cysteine loop was modeled based on the reported cysteine-loop sequence (Sterky *et al*, 2017), and glycosylation of the stalk region was modeled based on predicted *O*-linked glycosylation sites (Steentoft *et al*, 2013). Note the opposite effects on binding affinities of inserts at SS2 and SS4 for the Nrxn1 LNS2-Nxph1 and Nrxn1β-Nlgn1 complexes, respectively.

As with the I401Q mutation in Nrxn1, the equivalent mutations in Nrxn2 and Nrxn3 LNS2$^{SS2−}$ domains (referred to as Nrxn2 LNS2$^{SS2−}$ (I401Q) and Nrxn3 LNS2$^{SS2−}$ (I401Q) for simplicity) also significantly reduced binding affinities for Nxph1 (Fig 2E). Specifically, the I401Q mutations decreased the binding affinities of Nxph1 to the Nrxn2 and Nrxn3 LNS2$^{SS2−}$ domains from the nanomolar to the micromolar range and from nanomolar to the millimolar range, respectively (Fig 2E; Table 2). These binding data suggest that there

may be some isoform specificity to the interaction of Nxph1 with Nrxn1, Nrxn2, and Nrxn3 that becomes apparent when the interaction is destabilized by the Nrxn I401Q mutation.

Binding of Nxph3 to Nrxn1–3 LNS2$^{SS2−}$ was also tested. Similar to Nxph1, Nxph3 binds to the Nrxn1, Nrxn2, and Nrxn3 LNS2$^{SS2−}$ domains with nanomolar affinities (Fig 5B; Table 2). Furthermore, using BLI, the I401Q mutation in Nrxn1–3 LNS2$^{SS2−}$ showed reduced binding of Nxph3, increasing the $K_D$ of Nxph3 to Nrxn1–3

from the nanomolar to the micromolar range (Fig EV5B; Table 2). Comparison of Nxph1 and Nxph3 binding to wild-type and I401Q mutant Nrxn1–3 reveals that the reduction in binding by the I401Q mutation is not as large for Nxph3 as it is for Nxph1 (Fig EV5B; Table 2). These binding data suggest that isoform specificity may exist for Nxph1 and Nxph3 binding to neurexins. Unfortunately, we were unable to produce recombinant Nxph2 and Nxph4 alone or in a complex with Nrxn1 LNS2$^{SS2-}$, precluding quantitative measurements of interactions between neurexins and these proteins.

In addition to quantitative binding assays, we performed qualitative confocal fluorescent imaging assays of HEK293T cells co-expressing membrane-bound Nrxn1 LNS2$^{SS2-}$ or Nrxn1 LNS2$^{SS2-}$ (I401Q) with secreted Nxph1–4 (Fig EV5A). These co-expression data show that the I401Q mutation in Nrxn1 LNS2$^{SS2-}$ reduces the localization of Nxph1–3 with Nrxn1 LNS2$^{SS2-}$ (Fig EV5A). Unfortunately, Nxph4 did not express well in these cells.

Together, these BLI and imaging data confirm that mouse neurexins bind to Nxph1, Nxph2, and Nxph3, consistent with the conservation of the Nxph1-binding site in vertebrate neurexins.

# Discussion

Here, we describe high-resolution structures of a neurexophilin and analyzed its interaction with neurexins by crystallization of Nxph1 in a complex with the LNS2$^{SS2-}$ and the LNS2$^{SS2A+}$ domains of Nrxn1. These analyses reveal new atomic-resolution insights into how alternative splicing of a neurexin shapes its interactions with a ligand. The structures reveal a unique binding interface wherein the β-sandwiches of LNS2 and Nxph1 augment one another, essentially forming a single contiguous domain (Fig 1B). This heteromeric β-sandwich represents a previously unreported mode of a protein–protein interaction as assessed by PISA searches of all deposited structures in the Protein Data Bank (RCSB). The Nxph1-LNS2 complexes are formed by anti-parallel β-strand interactions as well as by additional specific hydrophobic and polar interactions that span the Nxph1-LNS2 β-sandwich (Figs 1B, 2A, and 3A; Appendix Fig S3C and D). Together, these interactions contribute to the strength of Nxph1-LNS2 binding, consistent with its nanomolar affinity (Figs 2E, 3E, and 5; Table 2). Furthermore, mutation of a key neurexin residue (I401) substantially reduced binding of Nxph1 and Nxph3 to the LNS2$^{SS2-}$ variants of all three neurexins (Figs 2E and EV5B).

## Effects of alternative splicing

Intriguingly, alternative splicing of the Nrxn1 LNS2 domains modulates the architecture of the Nxph1-LNS2 complexes. Alternative splicing alters the strength of neurexophilin binding to neurexin by remodeling the pattern of interactions at and near the interface between the two molecules. Specifically, extension of LNS2 $\beta_9$ by the SS2A insert (Figs 1B and 3A) increases the affinity of the Nxph1-LNS2 interaction by enhancing the interaction kinetics (Fig 3D; Table 2). This is accomplished by the addition of specific contacts at the Nxph1-LNS2 interface in the presence of SS2A as well as by increasing the number of stabilizing hydrophobic interactions nearby (Fig 3B). These structural features likely

explain why LNS2$^{SS2A+}$ and LNS2$^{SS2AB+}$ have higher affinities for Nxph1 than LNS2$^{SS2-}$ (Fig 3E). Mutation of a key interacting isoleucine residue in both Nrxn1 LNS2$^{SS2A+}$ and Nrxn1 LNS2$^{SS2AB+}$ also significantly reduced the binding affinity of Nxph1 for those domains (Fig 3E), suggesting that SS2 inserts augment the binding of LNS2 to Nxph1.

The effects of alternative splicing on the neurexophilin–neurexin interaction contrast with previous findings for a related synaptic protein interface. While LNS2 domains with SS2 inserts show higher affinity for Nxph1 than insert-free LNS2, the opposite is true for neurexin–neuroligin interactions, where Nrxn1β with a 30-amino acid insert in SS4 (β-Nrxn1$^{SS4+}$) has a lower affinity for neuroligins than β-Nrxn1$^{SS4-}$ (Boucard et al, 2005; Elegheert et al, 2017; Fig 6). In this case, the SS4 insert functions differently than SS2 inserts; the SS4 insert likely sterically interferes with binding of neuroligin to β-Nrxn1$^{SS4+}$ (Shen et al, 2008), whereas LNS2 with SS2A inserts— and likely SS2AB inserts—form additional interactions with Nxph1 not found between Nxph1 and LNS2$^{SS2-}$. However, in the case of LNS2 the role of alternative splicing is similar to that of cerebellin binding in the context of LNS6 alternative splicing, where cerebellin only binds LNS6 domains augmented with a SS4 insert (Uemura et al, 2010; Matsuda & Yuzaki, 2011; Elegheert et al, 2016). Unfortunately, there is no high-resolution structure available of the cerebellin-LNS6$^{SS4+}$ complex to reveal how SS4 affects the interaction. Nevertheless, low-resolution negative stain electron microscopy data suggest that LNS6$^{SS4+}$ interacts with the flexible cysteine-rich region in cerebellin-1 (Cheng et al, 2016; Elegheert et al, 2016) and likely cerebellin-4 (Zhong et al, 2017).

## General insights into neurexophilin architecture and conservation

These structural data enable an accurate definition of the domain structure of neurexophilins. Previous sequence analyses proposed that the mature fragment of neurexophilins consists of separate N- and C-terminal domains (Missler & Südhof, 1998; Missler et al, 1998; Reissner et al, 2014), but the crystal structures presented here reveal that mature Nxph1 does not contain two independently folded domains. Thus, the primary structure of neurexophilins is redefined here as consisting of a signal peptide, pro-domain, polybasic cleavage site, and mature fragment. Nxph4 contains an extra sequence insert between $\beta_4$ and $\beta_5$, which presumably emerges as a large loop from the smaller face of the neurexophilin β-sandwich (Appendix Fig S2).

Key residues of the Nrxn1 LNS2-binding site in Nxph1 are highly conserved in all vertebrate neurexophilins (Fig EV3), as are the Nxph1-binding residues in neurexins (Fig EV2). Moreover, binding data show that Nxph1 binds to all neurexins with nanomolar affinity, as does Nxph3 (Fig 5). Importantly, Nxph1 binding is not only regulated by SS2 alternative splicing, but may also differ depending on which of the three neurexins is present (Fig 5); in other words, a particular neurexophilin may bind preferentially to a particular neurexin with additional fine-tuning achieved through SS2 alternative splicing. Although the observed differences in affinities are small, they could be amplified in the cellular context with multiple neurexins and neurexophilins interacting in a restricted space. This exciting possibility would imply that separate isoform-specific, neurexin–neurexophilin signaling pathways could be organized in

space. Future research will have to address this possibility by conducting systematic binding experiments of synaptic assemblies. Finally, conservation suggests that this tuning is a relatively recent evolutionary innovation. While the Nxph1-binding site is highly conserved in vertebrate LNS2, it is not present in invertebrate neurexin sequences (Fig EV2), suggesting that the neurexophilins may have evolved in vertebrates as neurexin ligands. The conservation of the neurexophilins in vertebrates is striking, and their absence from invertebrates may reflect an important role of the neurexophilins in vertebrate evolution.

### Concluding remarks

Nxph1 is likely an important regulator of synaptic transmission (Born *et al*, 2014), but what exactly Nxph1 and the other neurexophilins do and how they perform these functions remains to be defined. The high binding affinities of Nxph1 and Nxph3 for neurexins probably reflect a synaptic role for their interaction. Given that various neurexophilins are expressed in distinct populations of neurons in the central and enteric nervous systems (Földy *et al*, 2016; Tasic *et al*, 2016; Saunders *et al*, 2018; Zeisel *et al*, 2018), it is likely that neurexophilins contribute to the organization of synapses through neurexin binding. However, it is unclear, for example, whether neurexophilins function exclusively at the presynapse or whether they act at all of the output or input synapses of a neuron. The overall extent to which different neurexophilin and neurexin isoforms exhibit differential interactions is also unclear. Moreover, SS2 alternative splicing is tightly regulated temporally and spatially (Ullrich *et al*, 1995), but how that regulation relates to the function of neurexophilins and their possible competition with dystroglycan for binding (Reissner *et al*, 2014) remains to be investigated. The structural and biochemical data presented here can be used in structure-guided functional assays to further reveal the neurological significance of the neurexophilins.

From the Nxph1-LNS2 structures presented here, we observed a new protein interface architecture that is subject to regulation by alternative splicing. Our analysis of Nxph1-LNS2 interactions provides insight into how alternative splicing can regulate and fine-tune protein–protein interactions at the synapse. Furthermore, given the general role of alternative splicing in many important eukaryotic cellular contexts, our structural analysis is broadly relevant to understanding how diverse protein interactions can be modulated by alternative splicing.

## Materials and Methods

### Protein expression and purification for structural studies

For crystallographic studies, the mature form of rat Nxph1 (118–271) was expressed with an Igκ signal peptide fused to its N-terminus and an HRV 3C protease-cleavable double FLAG tag followed by a 6×-polyhistidine tag fused to its C-terminus. In this study, we focus on LNS2 domains, which are only present in the long-form Nrxnα and not the shorter Nrxnβ or Nrxn1γ; therefore, when we refer to Nrxn1, Nrxn2, and Nrxn3, we are referring to Nrxnα versions of Nrxn. The LNS2$^{SS2-}$ domain of mouse Nrxn1 (283–378; 394–480) was expressed with an Igκ signal peptide fused

to its N-terminus and an HRV 3C protease-cleavable Twin-Strep tag fused to its C-terminus. Note that Nrxn1 residues numbered here and referred to throughout the paper are in reference to the canonical UniProt mouse Nrxn1 Isoform 1a sequence: Q9CS84-1. In order to prevent artificial inter-molecular disulfides from forming between LNS2$^{SS2-}$, the LNS2$^{SS2-}$ C293A construct was made using the Quik-Change II Site-Directed Mutagenesis Kit (Agilent). We later found that inclusion of 50 nM TCEP in buffers could prevent inter-molecular disulfides from forming between LNS2, so we used 50 nM TCEP during purification of LNS2$^{SS2A+}$ [mouse Nrxn1 (283–386; 394–480)].

To obtain sufficient quantities of protein suitable for structural studies, constructs were cloned into the pEG BacMam vector and were co-expressed using the BacMam expression system in HEK293S GnTI- cells (Dukkipati *et al*, 2008; Goehring *et al*, 2014). Briefly, 293S cells were grown in FreeStyle Expression Media (Thermo Fisher) supplemented with 2% FBS at 37°C in 8% $CO_2$ to a density of 1–2 million cells per milliliter and co-transduced with Nxph1 and LNS2 viruses; approximately 8–24 h after transduction, 10 mM sodium butyrate was added to the expression cultures and the temperature reduced to 30°C. Proteins were expressed for approximately 72 h, and then, cell culture supernatants containing secreted proteins were harvested. Cell culture supernatants were concentrated, and buffer was exchanged into 20 mM Tris pH 8.0 with 150 mM NaCl (TBS 20/150 pH 8.0) and 10 mM imidazole at room temperature. The buffer-exchanged, recombinant protein-containing solution was loaded onto a 5-ml HisTrap (GE) using a ÄKTA Start (GE) at 4°C. After washing with 20 mM Tris pH 8.0 with 300 mM NaCl (TBS 20/300 pH 8.0) and 50 mM imidazole, a 5-ml StrepTrap column (GE) was connected in tandem to the HisTrap. Protein was then eluted off of the HisTrap and directly onto the StrepTrap using TBS 20/300 pH 8.0 and 300 mM imidazole (with or without 50 nM TCEP). Following a brief wash in TBS 20/300 pH 8.0 and 300 mM imidazole with or without 50 nM TCEP, protein complex bound to the StrepTrap was eluted from the column in TBS 20/300 pH 8.0 and 2.5 mM desthiobiotin with or without 50 nM TCEP (Appendix Fig S6).

Affinity-purified Nxph1-LNS2 complex was then dialyzed into 20 mM Tris pH 8.0 with 50 mM NaCl (TBS 20/50 pH 8.0; with or without 50 nM TCEP) in preparation for anion exchange chromatography, while affinity purification tags were removed using an HRV 3C protease digestion overnight at 4°C. Cleaved Nxph1-LNS2 complex was then injected onto a 1-ml HisTrap connected to a 1-ml StrepTrap in TBS 20/50 pH 8.0 and 20 mM imidazole (with or without 50 nM TCEP) in order to remove any remaining tagged protein. The flow-through from that step was then applied to a Mono Q 4.6/100 PE column (GE) in TBS 20/50 pH 8.0 with or without 50 nM TCEP; protein was then eluted with a 40 c.v. gradient of 20 mM Tris pH 8.0 with the concentration of NaCl ranging from 50 to 250 mM (TBS 20/50 pH 8.0 – TBS 20/250 pH 8.0). Fractions containing pure Nxph1-LNS2 were combined and then analyzed by SDS–PAGE and SEC-MALS.

### Crystallization and crystallographic data collection

For crystallization of Nxph1-LNS2 complexes, glycosylation of Nxph1 was prevented by mutating its *N*-linked glycosylation sites to aspartic acid (Nxph1$^{3ND}$). Nxph1$^{3ND}$-LNS2 complexes crystallized at

2–3 mg/ml in the 16% PEG 3350 with 2% Tacsimate pH 5.0 and 0.1 M sodium citrate tribasic condition from the PEG/Ion screen (Hampton Research). Nxph1$^{3ND}$-LNS2 crystals were cryoprotected in mother liquor containing 10% glycerol and 20% PEG 3350 and frozen in liquid $N_2$. For the Nxph1$^{3ND}$-LNS2$^{SS2-}$ crystals, data were collected at the Advanced Photon Source (APS) using the microfocus NE-CAT beamline 24-ID-C (Table 1). For the Nxph1$^{3ND}$-LNS2$^{SS2A+}$ crystals, data were collected at the Stanford Synchrotron Radiation Light Source (SSRL) using the microfocus beamline 12-2 (Table 1). For simplicity, we refer to Nxph1$^{3ND}$ simply as Nxph1 when discussing the crystal structures. We note, however, that all biochemical studies were performed with Nxph1 without the 3ND mutations, as we were unable to prepare well-behaved Nxph1$^{3ND}$ alone.

## Crystallographic data processing, structure determination, and refinement

Crystallographic datasets were indexed, integrated, and scaled using HLK2000 (Otwinowski & Minor, 1997). We first determined the structure of Nxph1 in a complex with LNS2$^{SS2-}$ using molecular replacement with a pre-existing LNS2 structure (2H0B; Sheckler *et al*, 2006) as a search model, with the LNS2 domain amounting to roughly half of the complex. Electron density for the Nxph1 molecule emerged in difference maps, and a model of its structure was built *ab initio* using phenix.autobuild followed by iterative manual building and refinement using Coot (Emsley *et al*, 2010) and automated refinement using phenix.refine (Adams *et al*, 2010). We determined the structure of Nxph1 in a complex with LNS2$^{SS2A+}$ using molecular replacement with the Nxph1-LNS2$^{SS2-}$ structure as a search model. The Nxph1-LNS2$^{SS2A+}$ structure was also refined with iterative manual building and refinement using Coot and automated refinement using phenix.refine. For the Nxph1-LNS2$^{SS2-}$ structure, we modeled the full Nrxn1 LNS2$^{SS2-}$ domain (residues 283–378; 394–480) and residues 119–180, 191–212, and 217–263 of Nxph1. For the Nxph1-LNS2$^{SS2A+}$ structure, we modeled residues 283–378, 383–386, and 394–480 of LNS2$^{SS2A+}$ and residues 119–180, 189–214, and 217–261 of Nxph1. The regions that were not modeled in the structures, including residues 379–382 of SS2A, had weak and disordered electron density.

## Analysis of crystallographic interfaces

Interface areas were calculated using PISA, and the search of the Protein Data Bank (RCSB) was performed using PISA (Krissinel & Henrick, 2007).

## Structural model figure preparation

Figures showing structural models were made using PyMOL (The PyMOL Molecular Graphics System, version 2.0 Schrödinger, LLC).

## Sequence alignments

Sequences were aligned using default parameters in CLC Main Workbench 7 and 9 (Qiagen), and secondary structures from the Nxph1-LNS2 structures were assigned to alignments using ESPript 3 (Robert & Gouet, 2014).

## Protein expression and purification for biochemical studies

Neurexin LNS2 mutants were made using the Q5 site-directed mutagenesis kit (NEB). For BLI studies, neurexins were expressed with an Igκ signal peptide fused to their N-terminus and an AviTag followed by hexahistidine tag on their C-terminus. For co-immunoprecipitation studies, neurexins were expressed with an Igκ signal peptide fused to their N-terminus and a Twin-Strep tag on their C-terminus. For both co-immunoprecipitation and BLI studies, neurexin constructs were transfected into 293F cells in small-scale (25–400 ml) cultures using PEI (Sigma) in a 3:1 PEI-to-DNA ratio (Aricescu *et al*, 2006) or using the Expi-293 expression system (Thermo Fisher); cultures were grown at 37°C in 8% $CO_2$. 10 mM sodium butyrate or enhancers were added to the cultures 8–24 h post-transfection. Approximately 72 h later, secreted neurexin proteins in the media were bound in batch to either Strep-Tactin resin (IBA Biosciences) or Ni-NTA resin (Qiagen) and purified using affinity chromatography. For co-immunoprecipitation studies, neurexins were purified using Strep-Tactin affinity chromatography (Schmidt *et al*, 2013). For BLI studies, neurexins were purified using Ni-NTA affinity chromatography, biotinylated enzymatically with BirA (Avidity) overnight, and further purified using a Superdex Increase 200 10/300 gel filtration column (GE).

For fluorescence anisotropy binding studies, Nrxn1 LNS2$^{SS2-}$-Twin-Strep and Nrxn1 LNS2$^{SS2A+}$-Twin-Strep were expressed in 293F cells using the BacMam expression system. Secreted LNS2 proteins were purified from media using Strep-Tactin affinity chromatography, and affinity tags were removed overnight using HRV 3C protease. LNS2 proteins were further purified using anion exchange chromatography and then labeled on a native thiol on cysteine residue 293 with Alexa Fluor™ 488 C5 Maleimide (Thermo Fisher) overnight in TBS 20/150 pH 8.0 with 50 nM TCEP.

Neurexophilins expressed poorly when transfected into 293F cells using PEI or ExpiFectamine. Therefore, the BacMam expression system was used to express Nxph1–4 in 293F cultures at 37°C and 8% $CO_2$. The same Nxph1 construct used for crystallographic studies (but without asparagine residues mutated) was used in all biochemical studies. Rat Nxph2 (108–261), rat Nxph3 (50–252), and rat Nxph4 (105–304) were expressed with an Igκ signal peptide fused to their N-terminus and an HRV 3C protease-cleavable double FLAG tag followed by a 6×-polyhistidine tag fused to their C-terminus. All Nxph proteins used in biochemical studies contained affinity tags and glycans. We were unable to produce well-behaved Nxph2 and Nxph4.

## SEC-MALS data collection

Data were collected using a Superdex 200 10/300 column at a flow rate of 0.5 ml/min in phosphate-buffered saline pH 7.4 (PBS pH 7.4; Sigma). UV absorption, light scattering, and differential refractometry data were recorded for protein elution profiles using Dawn Heleos-II and Optilab rEX instruments (Wyatt technology). Baselines were corrected with Astra 7.1.2 (Wyatt technology) using a BSA reference and processed with a differential refractive index value (dn/dc) value of 0.185.

## SEC-SAXS data collection and analysis

SEC-SAXS data were collected at SSRL beamline 4-2. Briefly, protein samples were injected onto a Superdex Increase s200 3.2/300

column (GE) using an UltiMate 3000 HPLC (Thermo Fisher), and SAXS data were collected with 1-s exposures every 5 s as the proteins eluted off of the column. Data were collected using beam energy of 11 keV and a PILATUS3 X 1 M detector (Dectris). Peak SEC-SAXS data points were selected for SAXS data analysis. Crystallographic models were fit to SAXS data using CRYSOL, and molecular weights were determined using PRIMUS in the ATSAS software suite (Franke *et al*, 2017).

### Co-IP binding assays

For Co-IP binding assays, 10 μg of LNS2 protein was mixed with 10 μg of Nxph1 in 20 mM HEPES pH 7.4 with 150 mM NaCl (HBS). These protein mixtures were incubated with M2 anti-FLAG magnetic beads (Sigma) for 15 min at room temperature. The beads were harvested using a magnetic tube rack and washed three times with HBS. Protein complexes were then eluted from the beads using 30 μl of 100 μg/μl 3 × FLAG peptide (Sigma), and eluates were filtered using spin columns. Input and elution fractions were run on Bio-Rad Criterion TGX Any kD gels in Tris/Glycine/SDS running buffer (Bio-Rad) and transferred to 0.2 μm nitrocellulose using a Trans-Blot Turbo Transfer System (Bio-Rad). Nitrocellulose membranes were blocked overnight in TBST with 3% milk and then probed with either (1:2,000) mouse M2 anti-FLAG antibody (Sigma) or (1:1,000) rabbit anti-Strep-tag II antibody (Abcam) for 2 h at room temp. After washing, nitrocellulose membranes were probed with IRDye 800CW Donkey anti-Mouse IgG (H + L) or IRDye 800CW Donkey anti-Rabbit IgG (H + L) secondary antibodies (LI-COR). After final washes, nitrocellulose membranes were imaged using a LI-COR Odyssey CLx imaging system.

### Co-expression and confocal imaging

Nrxn1 LNS2$^{SS2-}$ and Nrxn1 LNS2$^{SS2-}$ (I401Q) were expressed with an Igκ signal peptide fused to their N-terminus and an AviTag followed by a Myc tag and an HRV 3C protease-cleavable CD8 transmembrane domain fused to their C-terminus. The same Nxph1 and Nxph3 constructs with C-terminal FLAG tags used in the binding studies were used in these co-expression studies. Rat Nxph2 (108–261) and rat Nxph4 (105–304) were expressed with an Igκ signal peptide fused to their N-terminus and an HRV 3C protease-cleavable double FLAG tag followed by a 6×-polyhistidine tag fused to their C-terminus.

HEK293T cells were transfected using Lipofectamine 3000 (Thermo Fisher). After approximately 48 h, these cells were washed once with DPBS (Thermo Fisher), fixed with 4% paraformaldehyde, and then washed three times with DPBS. Fixed cells were then blocked with antibody dilution buffer containing 5% normal goat serum (Sigma) in DPBS (ADB) for 1 h. These cells were then incubated overnight without agitation at 4°C with 1:1,000 dilutions of mouse anti-FLAG M2 (Sigma) and polyclonal rabbit anti-cMyc (Sigma) in ADB. The cells were then washed three times with DPBS and incubated without agitation in 1:1,000 dilutions of goat anti-mouse Alexa Fluor 546 and goat anti-rabbit Alexa Fluor 647 in ADB for 1 h at room temp. Finally, stained cells were washed four times with DPBS and mounted on slides using DAPI Fluoromount-G Mounting Medium (Southern Biotech).

Cells were then imaged using a Nikon A1 Eclipse Ti confocal microscope using a 20× objective. Images were analyzed using NIS-Elements AR software (Nikon).

### CD spectroscopy

Proteins were diluted in TBS 20/150 pH 8.0 and 50 nM TCEP. CD spectra from 190 to 260 nm were collected using a 1.0 mm path length cell, 1.0 nm wavelength step, and 1.0-s averaging time at 25°C with an AVIV Circular Dichroism Spectrometer Model 202-01. For each protein sample, five technical replicates were recorded and then averaged.

### Biolayer interferometry

Kinetics data were collected with an Octet RED384 (Pall FortéBio) using a double reference subtraction method. Site-specific biotinylated neurexin constructs were immobilized onto streptavidin-coated tips and Nxph proteins associated and dissociated from the neurexin molecules. Kinetics experiments were performed at 25°C in freshly prepared PBS pH 7.4 with 50 nM TCEP and 1% BSA. At least four independent datasets were collected for each binding pair. Dissociation constants were calculated using the global fit method in the Octet Analysis software 9.0 (Pall FortéBio).

### Fluorescence anisotropy binding assays

Alexa 488-labeled LNS2 proteins were added at a concentration of 50 nM to a 1.5 × dilution series (from 0.07 to 20 μM) of Nxph1 in TBS 20/150 pH 8.0 and 50 nM TCEP with or without 2 mM $CaCl_2$ added. Alexa 488-labeled LNS2 proteins were also added at a concentration of 50 nM to wells containing TBS 20/150 pH 8.0 and 50 nM TCEP with or without 2 mM $CaCl_2$ added to serve as background controls. $Ca^{2+}$-free binding experiments were repeated as above but in the presence of 100 μM EDTA to determine if trace amounts of $Ca^{2+}$ in the buffer could affect Nxph1-LNS2 interactions. Plates containing these mixtures were mixed with shaking for 10 s three times and incubated at room temperature for 35 min before measuring fluorescence anisotropy using a BioTek Synergy 2 Multi-Mode Microplate Reader. Fluorescence polarization was calculated using BioTek Gen5 2.09 software, and background control values were subtracted from experimental Nxph1-LNS2 samples. Three independent experiments were performed for each binding pair and condition. Binding data were plotted in GraphPad Prism 7, and fits to data were made using a non-linear one-site total binding model with background constrained to zero.

### Statistics

For comparing binding affinities, two-tailed Welch's *t*-tests were used. *P*-values are indicated in the figures as: *$P < 0.05$, **$P < 0.01$, ***$P < 0.001$, and ****$P < 0.0001$. Crystallographic data and refinement statistics are provided in Table 1.

## Data availability

Processed biolayer interferometry traces are available on request. The Nxph1-LNS2$^{SS2-}$ and Nxph1-LNS2$^{SS2A+}$ structures and

diffraction data have been submitted to the Protein Data Bank (RCSB) [PDB IDs 6PNP (http://www.rcsb.org/pdb/explore/explore.do?structureId = 6pnp) and 6PNQ (http://www.rcsb.org/pdb/explore/explore.do?structureId = 6pnq)].

Expanded View for this article is available online.

## Acknowledgements

We thank the Brunger, Südhof, Garcia, and Weis laboratories for their support and input, Kang Shen for discussions, Justin Trotter for Nrxn3 constructs, Anna Khalaj for confocal image analysis advice, Lior Almagor for help with the fluorescence anisotropy binding experiments, Mark Stolowitz and the Canary Center at Stanford for use of the Octet RED384, and Tsutomu Matsui for help with SEC-SAXS data collection and analysis. The pEG BacMam vector was a gift from Eric Gouaux, and the Nxph1[3ND] construct was a gift from Markus Missler and Carsten Reissner. S.C.W. is supported by the Stanford Genome Training Program. Crystallographic data were collected with support from Raj Rajashankar and APS NE-CAT beamline 24-ID-E staff and the staff of SSRL beamline 12-2. This work is based upon research conducted at the Northeastern Collaborative Access Team beamlines, which are funded by the National Institute of General Medical Sciences from the National Institutes of Health (P30 GM124165). The PILATUS 6M detector on 24-ID-C beamline is funded by a NIH-ORIP HEI grant (S10 RR029205). This research used resources of the Advanced Photon Source, a U.S. Department of Energy (DOE) Office of Science User Facility operated for the DOE Office of Science by Argonne National Laboratory under Contract No. DE-AC02-06CH11357. SEC-SAXS experiments were performed at SSRL beamline 4-2. Use of the Stanford Synchrotron Radiation Lightsource, SLAC National Accelerator Laboratory, is supported by the U.S. Department of Energy, Office of Science, Office of Basic Energy Sciences, under Contract No. DE-AC02-76SF00515. The SSRL Structural Molecular Biology Program is supported by the DOE Office of Biological and Environmental Research and by the National Institutes of Health, National Institute of General Medical Sciences (including P41GM103393). The contents of this publication are solely the responsibility of the authors and do not necessarily represent the official views of NIGMS or NIH.

## Author contributions

SCW, TCS, and ATB conceived of the project and designed experiments. SCW, KIW, and QZ collected crystallographic data. SCW, KIW, QZ, and ATB determined and analyzed crystal structures. SCW generated constructs, purified all proteins, crystallized proteins, collected SEC-SAXS data, performed all binding assays, and performed confocal imaging experiments. RAP assisted SCW with SEC-MALS and CD data collection. UBC labeled proteins for fluorescence anisotropy binding experiments. SCW, KIW, TCS, and ATB wrote the paper with input from all authors.

## Conflict of interest

The authors declare that they have no conflict of interest.

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
