## [Review Process File · The EMBO Journal]

Structures of neurexophilin-neurexin complexes reveal a regulatory mechanism of alternative splicing

Steven C. Wilson, K. Ian White, Qiangjun Zhou, Richard A. Pfuetzner, Ucheor B. Choi, Thomas C. Südhof, and Axel T. Brunger

Review timeline:	Submission date:	21st Jan 2019
	Editorial Decision:	14th Mar 2019
	Editorial correspondence:	21st Mar 2019
	Revision received:	3rd Jul 2019
	Editorial Decision:	21st Aug 2019
	Revision received:	29th Aug 2019
	Accepted:	30th Aug 2019

Editor: Karin Dumstrei

Transaction Report:

1st Editorial Decision

21st Jan 2019

Thank you for submitting your manuscript to The EMBO Journal. I am sorry for the slight delay in getting back to you with a decision, but I was waiting for the third referee to return his/her report. I have still not received the last report and don't think that I will do so at this stage. I will therefore go ahead with the two referee reports on hand.

As you can see below, both referees appreciate the reported findings and support publication here. They raise a number of different issues that I would like to ask you to address in a revised version. The concerns raised are reasonable and I anticipate that you should be able to respond to them in a good way. Let me know if we need to discuss anything further.

REFeree REPORTS:

Referee #1:

Neurexins play an important role in synapse formation, using alternative splicing to engage several ligands. Little is known how alternative splicing affects ligand binding to neurexin, and in this study White et al. present crystal structures of neurexin/neurexophilin complexes for two splice variants. The structures show that the ligand binding mode is not much affected between the splice variants, despite an insertion of 8 residues at the ligand binding interface. Complementary binding assays show that the spliced insert increases ligand binding, and that other neurexins and neurexophilins use a similar binding mode. This manuscript provides first mechanical insight into the role of neurexin splicing in ligand binding, and I enthusiastically support publication in EMBO Journal.

I have one major concern regarding the interpretation of the oligomeric state of neurexophilin alone, which should be straightforward to address. The analysis, presentation and interpretation of the

structural differences between splice variants should be significantly improved, and some of the binding data are presented in a redundant manner.

Please find below a detailed criticism of the manuscript:

1. It is striking that the crystal forms for the Nrxn1 LNS SS2-/Nxph1 and Nrxn1 LNS SS2A+/Nxph1 complexes are so isomorphous. This illustrates that the ligand docking mode is very similar for both isoforms, but it also indicates that other molecular interfaces might stabilize crystal packing. The authors show in Figure S6 that the ligand Nxph1 itself forms a homodimer in the crystal, although the buried surface area is much lower than for the Nrxn1/Nxph1 interface. The authors conclude that there might be a weak Nxph1 dimer when Neurexin is absent, but the SEC-MALS data presented in Figure S1 suggest a mixture of dimer and tetramer. The MW fits for some of the SEC-MALS curves in Figure S1 are very poor (particularly for panels D, E & G). There is a simple explanation why they are poor at least for panels E and G showing Nxph1 and Nxph3. It seems the authors force to fit a dimer, where there is really a mixture of dimer and tetramer. Please note that both Nxph1 and Nxph3 seem to be oligomeric in solution, and no monomer is observed. Interestingly, Nxph1 becomes monomeric when bound to Neurexin. This suggests that there might be a competition between Nxph1 oligomerization and neurexin binding. Possibly, Nxph1 forms a homodimer at the same beta sandwich interface that is used for neurexin binding. The tetramer then occurs through the secondary homodimerization mode observed in the crystal structure. The oligomerization of the ligand observed in SEC-MALS affects the model presented in Figure 6. Since the SEC-MALS data are ambiguous, the authors should verify the oligomeric state of Nxph1 alone or in complex with Nrxn1 using a more analytical method such as SEC-SAXS or native mass spectrometry.

2. Although the crystal forms are isomorphous, there is a large difference in the overall atomic displacement between the splice variants. For some reason, the complex with the spliced insert is more rigid (given the low average B factor 35 as reported in Table I), whereas the complex lacking the insert has a very high overall B factor (76) that is not commensurate with the high resolution of the structure. The authors use an inhouse program to look at positional differences between the two complexes, after the normalization of the B factors. To me, it is quite surprising to see that the displacement seem to occur mostly along the center of the complex (Figure S8). Given the difference in B factors between the two complexes, I wonder whether the normalization is valid, but the authors give no details on their method other than two references. The comparison of the differences however small in the docking modes of the two isoforms is central to the paper, and requires a much more elaborate and careful analysis. The authors should provide buried surface area calculations for both isoforms, and show how much the 8 residue insert contributes in this respect. They should provide simple RMSD values for the superimposed complexes and the individual molecules. They should present an analysis of the tensors obtained from the TLS refinement, and maybe do a more elaborate TLS refinement with more than two TLS groups. If they insist to use their algorithm, they should compare the results to output from other similar programs available, such as Rapido or iPBA.

3. The authors should also do a better job to represent the local structural differences between the isoforms. Figure 3A is said to show a superimposition, but it seems to represent a single structure (and unlabeled side chains, whose origin is unclear). The differences maps presented in Figure 3B are quite convincing in showing that the structural differences are well-resolved in the electron density. But they do not illustrate the actual structural differences that matter. Why does the I401Q mutant affect ligand binding to the truncated isoform SS- more than to the SSA+ isoform? It would be nice if this were explained by showing local structural differences around I401Q, including the 8 residue extension.

4. Several panels in the main Figures could be designated for assignment to the supplement, besides Figure 3B. Table II lists all binding data between the different neurexins, their isoforms and different Nxph's. Yet, they are presented again in Figure 2F (where I think the numbers are swapped for Nrxn1 and Nrxn2, according to the same values presented in Table II), Figure 3 E and F (where it would be more comprehensive if the SS- data were also included despite its redundancy), and the entire Figure 5. All these figure panels are redundant, and unnecessary.

5. Based on a PISA server analysis that did not give any results, the authors claim that they have determined a new mode of heterodimerization: an anti-parallel beta-sandwich extension. Without attempting a very thorough analysis, I can come up with several examples that would have a similar binding topology, namely the classical T cell receptor and FAB ectodomains. If the authors were to insist on claiming the uniqueness of their binding mode, they should give a much more detailed

analysis on how Pisa was used, and why the binding mode is different from other similar heterodimers (such as for instance formed between Ig domains). In general, all the claims in the manuscript of being 'first' are in my opinion of poor taste and should be rewritten.

6. Instead, it would be useful to get an indication how unique the neurophilin fold is. This could be done with a simple DALI analysis.

7. The authors also assess whether calcium affects ligand binding (Figure 4). Is calcium actually observed in the crystal structures presented here? For a thorough analysis of calcium binding, neurexin should first be depleted from calcium for instance by treatment with EDTA or another metal chelator. Similar binding measurements should be done in the presence of a metal chelator, and with pretreated neurexin.

Minor issues:

1. The cartoon in Figure 1 shows probably unintended differences in secondary structure between the invariant parts of the two structures. Most strikingly: the b2 strand on LNS2 (in blue).

2. The TAP scheme presented in Figure S9 lacks the reverse purification step to get rid of the affinity tagged protein.

3. The scheme in Figure 6 begs the question if there is any interaction between the stalk of Neuroligin and Neurexophilin.

4. In the material and methods section, the description of the constructs is very confusing: "Nxph1 (118-271) was expressed as an N-terminal fusion to a HRV 3C protease-cleavable double FLAG tag followed by a 6×-polyhistidine tag using an Igk artificial signal peptide sequence." Sounds like the signal peptide is coming last, at the C-terminus. Please rewrite.

5. Please list manufacturers for a number of the materials such as Ni-NTA, AKTA, Strep columns, etc.

6. Please write in full the TBS buffer compositions the first time they occur. I can sort of understand that TBS 20/300 means 20 mM Tris and 300 mM NaCl, but better write in full.

7. It seems the crystals were frozen in a bufferless solution (Page 18)?

Referee #2:

Wilson and colleagues present a tidy and high quality paper on the biophysical/structural mechanism of neurexophilin-neurexin interaction. It includes a beautiful new complex crystal structure. This work fits in nicely with previous structural work in the neurexin field. It also highlights how insertion of an alternatively spliced sequence in Nrnx somewhat stabilises the interaction and thus modulates the binding affinity by ~6-fold.

Somewhat lacking is a careful discussion (and ideally some insights into the functional significance) of this interaction. For example, the structural data should enable the design of Nrnx and Nxphn mutants that have lost all binding affinity. These would be powerful tools in addition to the published I401Q, which appears to still bind with measurable affinity (eg see Table II and Fig 5F). Ideally, such mutants would then be tested in a functional assay. The Sudhof lab should be able to contribute with an elegant neuronal assay. At a minimum, the simple cell surface co-expression test described in Neupert et al 2015 could be repeated, comparing new mutants and wild type. That paper should be included in the discussion.

Small comments:

Fig 6: This is a beautiful figure, but it would help if the same view could be provided also with the structures shown as cartoon/ribbon, and all domains labelled.

Sup Fig 7A. Could the results from the pull-downs please be discussed in more detail? If there is an interesting new mutant in the panel that the authors suggest would be useful for future work then the affinity should be assessed with a proper binding assay (BLI or SPR or ITC...).

I have finally received the third referee report - provided below. The referee brings up a number of good points that should be fairly easy to address. I would therefore like to ask you to address the comments below in a revised version.

REFeree REPORTS:

Referee #3

Report for Author:

Numerous studies have sought to establish the molecular mechanisms by which the presynaptic cell adhesion molecule neuexin (Nrx) interacts with ligands to regulate synapse function. One such ligand is neuexophilin (Nxph); a family of small, neuropeptide-like secreted proteins. While previous studies (Missler & Sudhof, 1998, Missler et al., 1999, Reissner et al., 2014) had established that the alpha-Nrx-Nxph interaction is mediated via the Nrx LNS2 domain, in an apparently splice-independent manner, the precise molecular details of this interaction were unknown. Here, the authors provide structural information for the interaction of the alpha Nrx1 LNS2 domain with Nxph1. The major finding of this study is that Nxph1 interacts with Nrx LNS2 (+/-SS2) via an extension of the jelly-roll beta-sandwich of LNS2 - a unique observation when compared with other structures in the PDB. On the basis of these structures, the authors perform biophysical analyses to establish that the SS2

insert enhances binding affinity and identify a key role for the alpha-Nrx I401 residue in the Nxph-Nrx interaction. While the study will be of interest to those studying the structural aspects of Nrx-ligand interactions, it does not go on to provide new insight into the physiological significance of Nrx-Nxph interactions. It would be of interest to assess how perturbing this interaction (on the basis of the structural data presented) affects synaptic function for example. Nevertheless, I support publication of this manuscript in the EMBO Journal, on the basis of structural novelty and comprehensive biophysical validation.

Suggestions for manuscript improvement:

1. While the authors propose a mechanism of alpha Nrx isoform-specific signalling, this is only revealed through introducing a key-destabilizing mutant (I401). Is mutation of this residue a known genetic variant, which results in disease/disorder states? Without functional data to elaborate on proposed isoform-specific events, it would be of use to at least establish whether the effects of I401 extend to the aNrx-Nxph3 interaction (through the biophysical methods used). This would be feasible given that purified Nxph3 is reportedly well behaved.

2. Nxph is a glycoprotein (with 3 N-linked glycans). While the authors state in the Methods that they used the 3ND mutant (Reissner et al., 2014) for obtaining well-diffracting crystals this is not stated in the main text. This is an important point and should be included in the main text. Notably, the authors use the fully glycosylated proteins for binding studies. It might be of interest to carry out a selection of the binding experiments with the the 3ND mutant, especially given the proposed role of Nxph glycans in strengthening the interaction with alpha-Nrx (Reissner et al., 2014). At the very least the authors should acknowledge and comment on the potential roles of Nxph glycosylation.

3. On the basis of mapping per-residue differences, the authors propose that Nxph1 binding may be able to induce allosteric effects at "distant interfaces with other possible partners". This begs the question of whether the authors assessed feasibility of generating purified full-length alpha Nrx (comprising LNS1-6) to test binding of Nxph1 and how this might affect binding of other partners, e.g. neuroligin, LRRTM, dystroglycan. Biochemical & biophysical experiments to assess this would bolster the significance of this study in the absence of functional studies.

4. Neuroligins are dimers, and the model shown in Figure 6 should take this into account. It would be a more accurate description of reality, and also make the figure more informative.

Minor comments:

Page 7. While the alignments Sup Figs 2 and 4 are extensive, some of the details referred to in the text are lost. The authors might consider truncating to include fewer species. Additionally, the Nxph-binding site is not conserved in invertebrate neurexins however invertebrate sequences are not included in the alignment unfortunately...

Page 8. Fig 2A - The Nxph residues are difficult to see with the yellow surface. The authors might consider adjusting the surface transparency.

Fig2C - Please specify the contour level for the electron density map.

Page 10. Fig 3A-C. While it is good to see the electron density maps, it is difficult to see the structural details/comparisons of alternative splicing. The authors might consider enlarging these panels, or moving density maps to supplementary information.

Page 12. What accounts for the discrepancy between the Kd reported by BLI and fluorescence anisotropy?

Page 15. A number of studies have assessed the LNS6-SS4+-Cerebellin complex using (low resolution) structural approaches. This sentence should be clarified and the relevant references cited - Elegheert et al., Science (2016), Cheng et al., Structure (2016) & Zhong et al. Cell Reports (2017).

Page 31 (Supplementary figure S3). A label for the beta10 strand in left panel could be included.

1st Revision - authors' response

3rd Jul 2019

We are grateful for the constructive comments and suggestions by reviewers. Please find our responses to the comments below (blue font).

Reviewer comments:

Referee #1:

Neurexins play an important role in synapse formation, using alternative splicing to engage several ligands. Little is known how alternative splicing affects ligand binding to neurexin, and in this study White et al. present crystal structures of neurexin/neurexophilin complexes for two splice variants. The structures show that the ligand binding mode is not much affected between the splice variants, despite an insertion of 8 residues at the ligand binding interface. Complementary binding assays show that the spliced insert increases ligand binding, and that other neurexins and neurexophilins use a similar binding mode. This manuscript provides first mechanical insight into the role of neurexin splicing in ligand binding, and I enthusiastically support publication in EMBO Journal.

I have one major concern regarding the interpretation of the oligomeric state of neurexophilin alone, which should be straightforward to address. The analysis, presentation and interpretation of the structural differences between splice variants should be significantly improved, and some of the binding data are presented in a redundant manner.

Please find below a detailed criticism of the manuscript:

1. It is striking that the crystal forms for the Nrnx1 LNS SS2-/Nxph1 and Nrnx1 LNS SS2A+/Nxph1 complexes are so isomorphous. This illustrates that the ligand docking mode is very similar for both isoforms, but it also indicates that other molecular interfaces might stabilize crystal packing. The authors show in Figure S6 that the ligand Nxph1 itself forms a homodimer in the crystal, although the buried surface area is much lower than for the Nrnx1/Nxph1 interface. The authors conclude that there might be a weak Nxph1 dimer when Neurexin is absent, but the SEC-MALS data presented in Figure S1 suggest a mixture of dimer and tetramer. The MW fits for some of the SEC-MALS curves in Figure S1 are very poor (particularly for panels D, E & G). There is a simple explanation why

they are poor at least for panels E and G showing Nxph1 and Nxph3. It seems the authors force to fit a dimer, where there is really a mixture of dimer and tetramer. Please note that both Nxph1 and Nxph3 seem to be oligomeric in solution, and no monomer is observed. Interestingly, Nxph1 becomes monomeric when bound to Neurexin. This suggests that there might be a competition between Nxph1 oligomerization and neurexin binding. Possibly, Nxph1 forms a homodimer at the same beta sandwich interface that is used for neurexin binding. The tetramer then occurs through the secondary homodimerization mode observed in the crystal structure. The oligomerization of the ligand observed in SEC-MALS affects the model presented in Figure 6. Since the SEC-MALS data are ambiguous, the authors should verify the oligomeric state of Nxph1 alone or in complex with Nrnx1 using a more analytical method such as SEC-SAXS or native mass spectrometry.

We agree that the MW fits for the SEC-MALS data for Nxph1 and Nxph3 in panels E and G, respectively, are not ideal, but are the best that were achievable. As the reviewer suggested, we verified the oligomerization states of Nxph1 alone and in complex with neurexin using SEC-SAXS at SSRL (Supplementary Figure S7). We obtained high quality SEC-SAXS data for Nxph1, the Nxph1^{3ND}-Nrnx1 LNS2^{SS2A+} complex, and Nrnx1 LNS2^{SS2A+} alone. The SEC-SAXS peaks match our SEC-MALS results well. From these data, we used the ATSAS software suite to determine the molecular weights of the species present in the major Nxph1 peak as well as the higher molecular weight shoulder of that peak. The major peak for Nxph1 contains species corresponding to Nxph1 dimer, and the shoulder of that peak contains species that correspond to Nxph1 tetramer. This analysis confirms that Nxph1 exists in dimer-tetramer equilibrium when alone in solution.

We were able to fit our crystal structures to the SEC-SAXS data for the Nxph1^{3ND}-Nrnx1 LNS2^{SS2A+} complex and Nrnx1 LNS2^{SS2A+} using the program CRY SOL from EMBL. However, due to glycosylation of our Nxph1 sample, we were unable to obtain a good fit of our non-glycosylated Nxph1 dimer model to the Nxph1 SAXS data. We attempted to prepare Nxph1^{3ND} and EndoF1 deglycosylated Nxph1 samples, but in both cases, the samples aggregated irreversibly. In any case, the SEC-SAXS also suggest that there exist both dimer and tetrameric species of Nxph1 alone. The SEC-SAXS data confirm that Nxph1 in complex with Nrnx1 LNS2 forms a 1:1 heterodimer in solution.

We agree with the reviewer that it is possible that the Nrnx-Nxph1 interface observed in our structure may also be a Nxph1 homodimerization interface which is disrupted upon Nrnx binding, and that the Nxph1-Nxph1 interface in our crystal structure could be another interface involved in tetramerization. Because the precise nature of the dimer and tetramer is uncertain, we have changed the text accordingly.

2. Although the crystal forms are isomorphous, there is a large difference in the overall atomic displacement between the splice variants. For some reason, the complex with the spliced insert is more rigid (given the low average B factor 35 as reported in Table I), whereas the complex lacking the insert has a very high overall B factor (76) that is not commensurate with the high resolution of the structure. The authors use an inhouse program to look at positional differences between the two complexes, after the normalization of the B factors. To me, it is quite surprising to see that the displacement seem to occur mostly along the center of the complex (Figure S8). Given the difference in B factors between the two complexes, I wonder whether the normalization is valid, but the authors give no details on their method other than two references. The comparison of the differences however small in the docking modes of the two isoforms is central to the paper, and requires a much more elaborate and careful analysis. The authors should provide buried surface area calculations for both isoforms, and show how much the 8 residue insert contributes in this respect. They should provide simple RMSD values for the superimposed complexes and the individual molecules. They should present an analysis of the tensors obtained from the TLS refinement, and maybe do a more elaborate TLS refinement with more than two TLS groups. If they insist to use their algorithm, they should compare the results to output from other similar programs available, such as Rapido or iPBA.

Because the analysis of the positional differences in the B-factor normalized structures is not essential to the conclusions of this work, we have removed it, and instead have provided superpositions of the complexes and provided the RMSD value (Figure 3A). We also feel that a more sophisticated analysis of differences in TLS groups would not provide new insights into the system.

We have added the buried surface area calculations for the interface in both structures to the text. In addition, we state that two residues of the SS2A inserts H384 and M386 have buried surface areas of approximately 13 \AA^2 and 12 \AA^2 , respectively.

3. The authors should also do a better job to represent the local structural differences between the isoforms. Figure 3A is said to show a superimposition, but it seems to represent a single structure (and unlabeled side chains, whose origin is unclear). The differences maps presented in Figure 3B are quite convincing in showing that the structural differences are well-resolved in the electron density. But they do not illustrate the actual structural differences that matter. Why does the I401Q mutant affect ligand binding to the truncated isoform SS- more than to the SSA+ isoform? It would be nice if this were explained by showing local structural differences around I401Q, including the 8 residue extension.

We have added an image showing an overall view of the aligned splice-variant structures to Figure 3A. We also better labeled the side chains in Figure 3A. We enlarged the panels in Figure 3B. In the enlarged panel on the left, the electron density associated with I401 suggests that there are no differences between the structures at this residue position. The major differences between the structures are near the SS2 insert site. The structures are essentially the same near I401 but differ considerably near SS2, where the SS2A insert increases binding affinity by conferring stabilizing interactions. We propose that the I401Q mutation has a greater effect on binding affinity in the complex lacking the SS2 insert because binding of Nxph1 to Nrnx is simply generally less stable when inserts in SS2 are absent.

4. Several panels in the main Figures could be designated for assignment to the supplement, besides Figure 3B. Table II lists all binding data between the different neurexins, their isoforms and different Nxph's. Yet, they are presented again in Figure 2F (where I think the numbers are swapped for Nrnx1 and Nrnx2, according to the same values presented in Table II), Figure 3 E and F (where it would be more comprehensive if the SS- data were also included despite its redundancy), and the entire Figure 5. All these figure panels are redundant, and unnecessary.

While the figures depicting the binding affinity measurements are to some degree redundant with each other and Table II, we believe that they are nevertheless valuable for the reader. We separated the binding data figures in order to simplify their comparison and to emphasize particular points while maintaining the flow and narrative of the paper, introducing some redundancy. We have combined the binding data in Figures 3E and 3F into one panel.

5. Based on a PISA server analysis that did not give any results, the authors claim that they have determined a new mode of heterodimerization: an anti-parallel beta-sandwich extension. Without attempting a very thorough analysis, I can come up with several examples that would have a similar binding topology, namely the classical T cell receptor and FAB ectodomains. If the authors were to insist on claiming the uniqueness of their binding mode, they should give a much more detailed analysis on how PISA was used, and why the binding mode is different from other similar heterodimers (such as for instance formed between Ig domains). In general, all the claims in the manuscript of being 'first' are in my opinion of poor taste and should be rewritten.

During the PISA interface analysis of our complexes, we took advantage of the ability of PISA to search the PDB for structures containing similar interfaces. Using the default interface search parameters (which look for interfaces with 70% similarity), the PISA interface search did not indicate any structures containing a similar interface to that seen in our structures. If the interface search criterion is lowered to 60% similarity, then the search turns up several structures of Galectin-7 dimers (i.e., 4uw4), which upon visual inspection do not adopt the beta-sandwich extension architecture seen in our structure. If the interface search criterion is lowered to 50% similarity, structures of agrin (i.e., 3v65), laminin (i.e., 1lhv), sex hormone-binding globulin (i.e., 1kdm), and T-cell receptors (i.e., 5u6q) emerge. Upon lowering the interface search criterion to 40% similarity, individual neurexin LNS structures emerge. All of those structures do contain beta-sandwich architectures. However, in none of those cases does visual inspection reveal a complex wherein the beta sandwiches of two molecules are essentially fused together, as is seen in our Nxph1-LNS2 complexes.

Complexes formed by the alpha and beta subunits of T-cell receptors or the heavy- and light-chains of Fab complexes contain beta-sandwich heteromers that interact, but they do not adopt

the same architecture observed in our structure. Instead, it appears that the faces of the beta-sandwiches in these complexes roll into one another with some heteromeric anti-parallel beta-strand interactions at the edges of the sandwiches. The structures of these complexes are quite different from that of the Nxph1-LNS2 complex.

The PISA search of the PDB for structures with a similar interface architecture as seen in our Nxph1-LNS2 complexes resulted in no significant matches. We would like to claim that the interface architecture in our complexes is unique to the PDB under the PISA search criteria. PDBePISA suggests that searches for interfaces similar to the query interfaces that do not produce results are indicative of a query interface unique to the PDB under the search criteria. We have added to the manuscript text an explanation that the default PISA search criteria were used to search the PDB for interfaces. Nevertheless, we have softened the text.

6. Instead, it would be useful to get an indication how unique the neurophilin fold is. This could be done with a simple DALI analysis.

We performed a DALI analysis of our Nxph1 structure. The top hit of the DALI search, with a z-score of 9.3, is a leukocidin subunit with 10% sequence identity to Nxph1. Other matches from the DALI search are AP-1 complex subunit gamma (an adaptin), ADP-Ribosylation factor-binding proteins, and alpha-hemolysin. We visually compared these structures with Nxph1 and found that despite low sequence identity, the topology of Nxph1 appears to be shared with that of the adaptin proteins identified by DALI. We have revised the text accordingly.

7. The authors also assess whether calcium affects ligand binding (Figure 4). Is calcium actually observed in the crystal structures presented here? For a thorough analysis of calcium binding, neurexin should first be depleted from calcium for instance by treatment with EDTA or another metal chelator. Similar binding measurements should be done in the presence of a metal chelator, and with pretreated neurexin.

We did not include calcium in any of our buffers and do not see any density corresponding to calcium in our crystal structures. The trace amounts of calcium present in ultrapure deionized water are unlikely to bind to LNS2 with strong occupancy given the low affinity (0.4 mM) of LNS2^{SS2-} for Ca²⁺ (Sheckler et al., *J. Biol. Chem.*, 2006). In any case, we re-performed the fluorescence polarization anisotropy binding assays without supplemental calcium using neurexin LNS2 pre-treated with 100 μM EDTA and buffer containing 100 μM EDTA. The presence of EDTA had no effect on the K_D values obtained for Nxph1 binding to Nrxn1 LNS2^{SS2-} or Nrxn1 LNS2^{SS2A+}. We have included a discussion of these results in the text and included the binding curves for these experiments in the supplement.

Minor issues:

1. The cartoon in Figure 1 shows probably unintended differences in secondary structure between the invariant parts of the two structures. Most strikingly: the b2 strand on LNS2 (in blue).

We reassigned the secondary structure of the Nxph1-Nrxn1 LNS2^{SS2-} structure to match that of the Nxph1-Nrxn1 LNS2^{SS2A+} structure.

2. The TAP scheme presented in Figure S9 lacks the reverse purification step to get rid of the affinity tagged protein.

We added this reverse purification step to the figure showing the tandem affinity purification scheme.

3. The scheme in Figure 6 begs the question if there is any interaction between the stalk of Neuroligin and Neurexophilin.

Thank you for pointing this out. We revised Figure 6 to show a supercomplex dimerized by Neuroligin.

4. In the material and methods section, the description of the constructs is very confusing: "Nxph1 (118-271) was expressed as an N-terminal fusion to a HRV 3C protease-cleavable double FLAG tag

followed by a 6×-polyhistidine tag using an Igκ artificial signal peptide sequence." Sounds like the signal peptide is coming last, at the C-terminus. Please rewrite.

We rewrote the construct descriptions in the material and methods section.

5. Please list manufacturers for a number of the materials such as Ni-NTA, AKTA, Strep columns, etc.

We rewrote this section to include all manufacturers of materials where appropriate.

6. Please write in full the TBS buffer compositions the first time they occur. I can sort of understand that TBS 20/300 means 20 mM Tris and 300 mM NaCl, but better write in full.

We rewrote the methods section so that the buffer compositions are written in full the first time that they occur.

7. It seems the crystals were frozen in a bufferless solution (Page 18)?

Thank you for pointing this out. We have noted in the text that the crystals were cryoprotected in mother liquor containing 10 % glycerol and 20 % PEG 3350.

Referee #2:

Wilson and colleagues present a tidy and high quality paper on the biophysical/structural mechanism of neurexophilin-neurexin interaction. It includes a beautiful new complex crystal structure. This work fits in nicely with previous structural work in the neurexin field. It also highlights how insertion of an alternatively spliced sequence in Nrnx somewhat stabilises the interaction and thus modulates the binding affinity by ~6-fold.

Somewhat lacking is a careful discussion (and ideally some insights into the functional significance) of this interaction. For example, the structural data should enable the design of Nrnx and Nxphn mutants that have lost all binding affinity. These would be powerful tools in addition to the published I401Q, which appears to still bind with measurable affinity (eg see Table II and Fig 5F). Ideally, such mutants would then be tested in a functional assay. The Sudhof lab should be able to contribute with an elegant neuronal assay. At a minimum, the simple cell surface co-expression test described in Neupert et al 2015 could be repeated, comparing new mutants and wild type. That paper should be included in the discussion.

We believe that the mutations presented in this work are sufficient for our conclusions. For example, the I401Q mutant decreases the affinity of Nxph1 to neurexin LNS2 domains by orders of magnitude. As the reviewer suggested, we performed a cell-surface binding assay to assess the effects of the neurexin I401Q mutation on binding to all neurexophilins in a co-expression context. The I401Q mutation decreases binding of secreted Nxph1–3 to membrane-localized neurexin, while poor expression of Nxph4 prevented analysis of Nxph4-neurexin binding. These data and a description of the assay are now included in the manuscript. We have also referenced Neupert et al 2015 in the main text.

Small comments:

Fig 6: This is a beautiful figure, but it would help if the same view could be provided also with the structures shown as cartoon/ribbon, and all domains labelled.

We thank the reviewer for the suggestion. However, we would prefer to leave it as a surface representation of the structure. This presentation required composing different models together in order to make a hypothetical full-length alpha-neurexin. During this process, some structural information is lost, and it would be misleading to show a more detailed representation of the proteins. We have revised the labeling to make the figure clearer.

Sup Fig 7A. Could the results from the pull-downs please be discussed in more detail? If there is an

interesting new mutant in the panel that the authors suggest would be useful for future work then the affinity should be assessed with a proper binding assay (BLI or SPR or ITC...).

We included a more detailed discussion of the new mutants in the text. We did not discuss the other point mutants in more detail because the binding energy of the interface is distributed across the interface, and I401 contributes the most energy of any one residue to this interface; any other single mutation will likely not have as great of an effect as I401Q.

Referee #3

Report for Author:

Numerous studies have sought to establish the molecular mechanisms by which the presynaptic cell adhesion molecule neurexin (Nrx) interacts with ligands to regulate synapse function. One such ligand is neuroligin (Nlgn); a family of small, neuropeptide-like secreted proteins. While previous studies (Missler & Südhof, 1998, Missler et al., 1999, Reissner et al., 2014) had established that the alpha-Nrx-Nlgn interaction is mediated via the Nrx LNS2 domain, in an apparently splice-independent manner, the precise molecular details of this interaction were unknown. Here, the authors provide structural information for the interaction of the alpha Nrx1 LNS2 domain with Nlgn1. The major finding of this study is that Nlgn1 interacts with Nrx LNS2 (+/-SS2) via an extension of the jelly-roll beta-sandwich of LNS2 - a unique observation when compared with other structures in the PDB. On the basis of these structures, the authors perform biophysical analyses to establish that the SS2

insert enhances binding affinity and identify a key role for the alpha-Nrx I401 residue in the Nlgn-Nrx interaction. While the study will be of interest to those studying the structural aspects of Nrx-ligand interactions, it does not go on to provide new insight into the physiological significance of Nrx-Nlgn interactions. It would be of interest to assess how perturbing this interaction (on the basis of the structural data presented) affects synaptic function for example. Nevertheless, I support publication of this manuscript in the EMBO Journal, on the basis of structural novelty and comprehensive biophysical validation.

Suggestions for manuscript improvement:

1. While the authors propose a mechanism of alpha Nrx isoform-specific signalling, this is only revealed through introducing a key-destabilizing mutant (I401). Is mutation of this residue a known genetic variant, which results in disease/disorder states? Without functional data to elaborate on proposed isoform-specific events, it would be of use to at least establish whether the effects of I401 extend to the alpha-Nrx-Nlgn3 interaction (through the biophysical methods used). This would be feasible given that purified Nlgn3 is reportedly well behaved.

Mutation of I401 is not known to be associated with any disease. In response to your comment, we performed BLI studies to examine the effects of the I401Q mutation in Nrxn1-3 LNS2^{SS2}- on Nlgn3 binding. We found that the I401Q mutation decreases the binding affinity of all neurexins to Nlgn3 but that this decrease is not as great as that seen with the I401Q mutants binding to Nlgn1. We have included these data and a discussion of them in the paper.

2. Nlgn is a glycoprotein (with 3 N-linked glycans). While the authors state in the Methods that they used the 3ND mutant (Reissner et al., 2014) for obtaining well-diffracting crystals this is not stated in the main text. This is an important point and should be included in the main text. Notably, the authors use the fully glycosylated proteins for binding studies. It might be of interest to carry out a selection of the binding experiments with the the 3ND mutant, especially given the proposed role of Nlgn glycans in strengthening the interaction with alpha-Nrx (Reissner et al., 2014). At the very least the authors should acknowledge and comment on the potential roles of Nlgn glycosylation.

We agree with the suggestion by the reviewer and have commented on the effects of glycosylation on crystallization and Nlgn1 binding to neurexin in the main text. We noted in the methods section that Nlgn1s used in all biochemical studies contained glycans. We would like to have conducted binding experiments using the Nlgn1^{3ND} mutant but despite numerous attempts, we were unable to produce well behaved Nlgn1^{3ND} protein on its own; that is, in all cases the Nlgn1^{3ND} protein suffered from gross aggregation issues as assessed by size-exclusion chromatography. Not being able to produce well-behaved Nlgn1^{3ND} protein, we attempted to deglycosylate Nlgn1 produced in HEK 293S GnTI- cells using Endoglycosidase F1. This sample could only be partially

deglycosylated (consistent with our observed partial deglycosylation of Nxp1 in complex with LNS2 during crystallization trials) and did not behave well either, again suffering from gross aggregation issues as assessed by size-exclusion chromatography.

3. On the basis of mapping per-residue differences, the authors propose that Nxp1 binding may be able to induce allosteric effects at "distant interfaces with other possible partners". This begs the question of whether the authors assessed feasibility of generating purified full-length alpha Nrx (comprising LNS1-6) to test binding of Nxp1 and how this might affect binding of other partners, e.g. neuroligin, LRRTM, dystroglycan. Biochemical & biophysical experiments to assess this would bolster the significance of this study in the absence of functional studies.

We tried to produce full-length alpha-neurexin using the BacMam expression system but did not produce high-yield preparations of full-length alpha-neurexin. We agree that it would be beneficial to do binding studies consisting of multiple synaptic proteins but feel that these studies are outside the scope of this paper.

4. Neuroligins are dimers, and the model shown in Figure 6 should take this into account. It would be a more accurate description of reality, and also make the figure more informative.

We originally made the model dimeric but decided to use a monomeric version for simplicity. We agree with your comment and remade the figure to take into account the neuroligin dimer.

Minor comments:

Page 7. While the alignments Sup Figs 2 and 4 are extensive, some of the details referred to in the text are lost. The authors might consider truncating to include fewer species. Additionally, the Nxp-binding site is not conserved in invertebrate neurexins however invertebrate sequences are not included in the alignment unfortunately...

We truncated the Nrxn and Nxp alignments and also included invertebrates in the Nrxn alignment.

Page 8. Fig 2A - The Nxp residues are difficult to see with the yellow surface. The authors might consider adjusting the surface transparency.

We adjusted the surface transparency for the Nxp1 surface.

Fig2C - Please specify the contour level for the electron density map.

Thank you for pointing this out. The contour level is 2σ .

Page 10. Fig 3A-C. While it is good to see the electron density maps, it is difficult to see the structural details/comparisons of alternative splicing. The authors might consider enlarging these panels, or moving density maps to supplementary information.

We enlarged the electron density maps and changed the left panel to show a larger region, which now spans the entire beta-sandwich interface of the complexes. In that left panel, the density for the I401 residue does not differ between the two structures.

Page 12. What accounts for the discrepancy between the K_d reported by BLI and fluorescence anisotropy?

Different binding assays often produce rather different K_D s. In any case, the calculated K_D s from the BLI and fluorescence anisotropy assays for each binding pair are within an order of magnitude of one another, and the trend in binding affinities with and without the SS2A insert present are the same. In both binding assays, the SS2A insert confers increased binding affinity of neurexin to Nxp1, validating the effect of the SS2A insert.

Page 15. A number of studies have assessed the LNS6-SS4+-Cerebellin complex using (low

resolution) structural approaches. This sentence should be clarified and the relevant references cited - Elegheert et al., Science (2016), Cheng et al., Structure (2016) & Zhong et al. Cell Reports (2017).

We added a discussion of the low-resolution negative stain EM data, which shows the Cerebellin-LNS6-SS4+ interaction. We also cited the appropriate references.

Page 31 (Supplementary figure S3). A label for the beta10 strand in left panel could be included.

We added this label. Thank you.

2nd Editorial Decision

21st Aug 2019

Thanks for submitting your revised manuscript to the EMBO Journal . Sorry for the slight delay in getting back to you with a decision, but I have now received the referee reports from referee #1 and 2. As you can see below, both referees appreciate the introduced changes and support publication here. Referee #1 has a few remaining concerns that can be addressed with appropriate text changes.

REFeree REPORTS:

Referee #1:

The manuscript is much improved, and most of my queries have been answered. I have two small issues left that I think should be addressed.

1. The authors have now done a DALI search to analyze the uniqueness of the Nxph1 domain structure. It seems to be very unique, but there is a discrepancy between the text in the rebuttal:

We performed a DALI analysis of our Nxph1 structure. The top hit of the DALI search, with a z-score of 9.3, is a leukocidin subunit with 10% sequence identity to Nxph1.

and what I found in the revised manuscript:

Analysis of the Nxph1 structure using DALI (Holm & Laakso, 2016) revealed that Nxph1 shares only very low-level structural similarity with a few proteins. Among these, the only proteins that have remote similarity to the Nxph1 topology are members of the adaptin family (sequence identity = 4%, RMSD = 3.3).

A sequence identity of 4% is random! Please include the structure from the rebuttal and mention its Z score, which is significant.

2. If the authors insist to show the poor MALLS fits in Fig EV1 panels D, E and G, please indicate in the legend that the fits are poor due to an oligomeric mixture.

Otherwise, I congratulate the authors on a wonderful study, and I look forward to see it in print!

Referee #2:

I am happy with the revisions made by the authors. The new structures presented are novel and interesting. The paper still lacks exploitation of these structural results to make significant advances in understanding the biology of the system, but the results are solid and I would be happy to see this published in EMBO.

2nd Revision - authors' response

29th Aug 2019

The authors performed the requested editorial changes.

3rd Editorial Decision

30th Aug 2019

Thank you for sending us your revised manuscript. I have now looked at everything and all looks good. I am therefore pleased to accept the manuscript for publication here.

Corresponding Author Name: Axel T. Brunger

Journal Submitted to: The EMBO Journal

Manuscript Number: EMBOJ-2019-101603R